# Dissecting Neural ODEs

**Stefano Massaroli**[*]
The University of Tokyo, DiffEqML
massaroli@robot.t.u-tokyo.ac.jp

**Michael Poli**[*]
KAIST, DiffEqML
poli_m@kaist.ac.kr

**Jinkyoo Park**
KAIST
jinkyoo.park@kaist.ac.kr

**Atsushi Yamashita**
The University of Tokyo
yamashita@robot.t.u-tokyo.ac.jp

**Hajime Asama**
The University of Tokyo
asama@robot.t.u-tokyo.ac.jp

## Abstract

Continuous deep learning architectures have recently re–emerged as *Neural Ordinary Differential Equations* (Neural ODEs). This infinite–depth approach theoretically bridges the gap between deep learning and dynamical systems, offering a novel perspective. However, deciphering the inner working of these models is still an open challenge, as most applications apply them as generic *black–box* modules. In this work we "open the box", further developing the continuous–depth formulation with the aim of clarifying the influence of several design choices on the underlying dynamics.

## 1 Introduction

Neural ODEs (Chen et al., 2018) represent the latest instance of continuous deep learning models, first developed in the context of continuous recurrent networks (Cohen and Grossberg, 1983). Since their introduction, research on Neural ODEs variants (Tzen and Raginsky, 2019; Jia and Benson, 2019; Zhang et al., 2019b; Yıldız et al., 2019; Poli et al., 2019) has progressed at a rapid pace. However, the search for concise explanations and experimental evaluations of novel architectures has left many fundamental questions unanswered.

In this work, we establish a general system–theoretic Neural ODE formulation (1) and dissect it into its core components; we analyze each of them separately, shining light on peculiar phenomena unique to the continuous deep learning paradigm. In particular, augmentation strategies are generalized beyond ANODEs (Dupont et al., 2019), and the novel concepts of *data–control* and *adaptive–depth* enriching (1) are showcased as effective approaches to learn maps such as *reflections* or *concentric annuli* without augmentation.

While explicit dependence on the depth–variable has been considered in the original formulation (Chen et al., 2018), a parameter depth–variance in continuous models has been overlooked. We provide a treatment in infinite–dimensional space required by the true *deep limit* of ResNets, the solution of which leads to a Neural ODE variant based on a spectral discretization.

**Neural Ordinary Differential Equation**

$$\begin{cases} \dot{\mathbf{z}} = f_{\boldsymbol{\theta}(s)}(s, \mathbf{x}, \mathbf{z}(s)) \\ \mathbf{z}(0) = h_x(\mathbf{x}) \\ \hat{\mathbf{y}}(s) = h_y(\mathbf{z}(s)) \end{cases} \quad s \in \mathcal{S} \quad (1)$$

| | | |
|---|---|---|
| Input | $\mathbf{x}$ | $\mathbb{R}^{n_x}$ |
| Output | $\hat{\mathbf{y}}$ | $\mathbb{R}^{n_y}$ |
| (Hidden) State | $\mathbf{z}$ | $\mathbb{R}^{n_z}$ |
| Parameters | $\theta(s)$ | $\mathbb{R}^{n_\theta}$ |
| Neural Vector Field | $f_{\theta(s)}$ | $\mathbb{R}^{n_z}$ |
| Input Network | $h_x$ | $\mathbb{R}^{n_x} \to \mathbb{R}^{n_z}$ |
| Output Network | $h_y$ | $\mathbb{R}^{n_z} \to \mathbb{R}^{n_y}$ |

---

[*]Equal contribution. Author order was decided by flipping a coin.

● **Depth–variance** Vanilla Neural ODEs (Chen et al., 2018) cannot be considered the deep limit of ResNets. We discuss the subtleties involved, uncovering a formal optimization problem in functional space as the price to pay for true depth–variance. Obtaining its solution leads to two novel variants of Neural ODEs: a Galërkin–inspired spectral discretization (GalNODE) and a piecewise–constant model. GalNODEs are showcased on a task involving a loss distributed on the depth–domain, requiring the introduction of a *generalized* version of the adjoint in (Chen et al., 2018).

● **Augmentation strategies** The augmentation idea of ANODEs (Dupont et al., 2019) is taken further and generalized to novel dynamical system–inspired and parameter efficient alternatives, relying on different choices of $h_x$ in (1). These approaches, which include *input–layer* and *higher–order* augmentation, are verified to be more effective than existing methods in terms of performance and parameter efficiency.

● **Beyond augmentation: data–control and adaptive–depth** We unveil that although important, augmentation is not always necessary in challenging tasks such as learning *reflections* or *concentric annuli* (Dupont et al., 2019). To start, we demonstrate that depth–varying vector fields alone are sufficient in dimensions greater than one. We then provide theoretical and empirical results motivating two novel Neural ODE paradigms: *adaptive–depth*, where the integration bound is itself determined by an auxiliary neural network, and *data–controlled*, where $f_{\theta(s)}$ is conditioned by the input data $\mathbf{x}$, allowing the ODE to learn a *family* of vector fields instead of a single one. Finally, we warn against input networks $h_x$ of the multilayer, nonlinear type, as these can make Neural ODE flows *superfluous*.

## 2 Continuous–Depth Models

**A general formulation** In the context of Neural ODEs we suppose to be given a stream of input–output data $\{(\mathbf{x}_k, \mathbf{y}_k)\}_{k \in \mathcal{K}}$ (where $\mathcal{K}$ is a linearly–ordered finite subset of $\mathbb{N}$). The inference of Neural ODEs is carried out by solving the *inital value problem* (IVP) (1), i.e.

$$\hat{\mathbf{y}}(S) = h_y \left( h_x(\mathbf{x}) + \int_{\mathcal{S}} f_{\theta(\tau)}(\tau, \mathbf{x}, \mathbf{z}(\tau)) \mathrm{d}\tau \right)$$

Our degree of freedom, other than $h_x$ and $h_y$, in the Neural ODE model is the choice of the parameter $\theta$ inside a given pre-specified class $\mathcal{W}$ of functions $\mathcal{S} \to \mathbb{R}^{n_\theta}$.

**Well–posedness** If $f_{\theta(s)}$ is Lipschitz, for each $\mathbf{x}_k$ the initial value problem in (1) admits a unique solution $\mathbf{z}$ defined in the whole $\mathcal{S}$. If this is the case, there is a mapping $\phi$ from $\mathcal{W} \times \mathbb{R}^{n_x}$ to the space of absolutely continuous functions $\mathcal{S} \mapsto \mathbb{R}^{n_z}$ such that $\mathbf{z}_k := \phi(\mathbf{x}_k, \theta)$ satisfies the ODE in (1). This in turn implies that, for all $k \in \mathcal{K}$, the map $(\theta, \mathbf{x}_k, s) \mapsto \gamma(s, \mathbf{x}_k, \theta) := h_y\big(\phi(\theta, \mathbf{x}_k)(s)\big)$ satisfies $\hat{\mathbf{y}} = \gamma(\theta, \mathbf{x}_k, s)$. For compactness, for any $s \in \mathcal{S}$, we denote $\phi(\theta, \mathbf{x}_k)(s)$ by $\phi_s(\theta, \mathbf{x}_k)$.

**Training: optimal control** (Chen et al., 2018) treated the training of constant–parameters Neural ODE (i.e. $\mathcal{W}$ is the space of constant functions) considering only *terminal* loss functions depending on the terminal state $\mathbf{z}(S)$. However, in the framework of Neural ODEs, the latent state evolves through a continuum of layers steering the model output $\hat{\mathbf{y}}(s)$ towards the label. It thus makes sense to introduce a loss function also distributed on the whole depth domain $\mathcal{S}$, e.g.

$$\ell := L(\mathbf{z}(S)) + \int_{\mathcal{S}} l(\tau, \mathbf{z}(\tau)) \mathrm{d}\tau \tag{2}$$

The training can be then cast into the *optimal control* (Pontryagin et al., 1962) problem

$$\begin{aligned} \min_{\theta \in \mathcal{W}} & \frac{1}{|\mathcal{K}|} \sum_{k \in \mathcal{K}} \ell_k \\ \text{subject to } & \dot{\mathbf{z}}(s) = f_{\theta(s)}\left(s, \mathbf{x}_k, \mathbf{z}(s)\right) \quad s \in \mathcal{S} \\ & \mathbf{z}(0) = h_x(\mathbf{x}_k), \ \ \hat{\mathbf{y}} = h_y(\mathbf{z}(s)) \end{aligned} \quad , \ \ \forall k \in \mathcal{K} \tag{3}$$

solved by gradient descent. Here, if $\theta$ is constant, the gradients can be computed with $\mathcal{O}(1)$ memory efficiency by generalizing the adjoint sensitivity method in (Chen et al., 2018).

**Proposition 1** (Generalized Adjoint Method). *Consider the loss function* (2). *Then,*

$$\frac{\mathrm{d}\ell}{\mathrm{d}\theta} = \int_{\mathcal{S}} \mathbf{a}^\top(\tau) \frac{\partial f_\theta}{\partial \theta} \mathrm{d}\tau \ \ \text{where } \mathbf{a}(s) \text{ satisfies} \ \begin{cases} \dot{\mathbf{a}}^\top(s) = -\mathbf{a}^\top \frac{\partial f_\theta}{\partial \mathbf{z}} - \frac{\partial l}{\partial \mathbf{z}} \\ \mathbf{a}^\top(S) = \frac{\partial L}{\partial \mathbf{z}(S)} \end{cases}$$

Appendix B contains additional insights on the choice of activation, training regularizers and approximation capabilities of Neural ODEs, along with a detailed derivation of the above result.

# 3 Depth-Variance: Infinite Dimensions for Infinite Layers

**Bring residual networks to the deep limit**  *Vanilla* Neural ODEs, as they appear in the original paper (Chen et al., 2018), cannot be fully considered the *deep limit* of ResNets. In fact, while each residual block is characterized by its own parameters vector $\theta_s$, the authors consider model $\dot{\mathbf{z}} = f_\theta(s, \mathbf{z}(s))$ where the depth variable $s$ enters in the dynamics *per se*[2] rather than in the map $s \mapsto \theta(s)$. The first attempt to pursue the true deep limit of ResNets is the *hypernetwork* approach of (Zhang et al., 2019b) where another neural network parametrizes the dynamics of $\theta(s)$.

However, this approach is not backed by any theoretical argument and it exhibits a considerable parameter inefficiency, as it generally scales polynomially in $n_\theta$. We adopt a different approach, setting out to tackle the problem theoretically in the general formulation. Here, we uncover an optimization problem in functional space, solved by a direct application of the adjoint sensitivity method in infinite-dimensions. We then introduce two parameter efficient depth–variant Neural ODE architectures based on the solution of such problem: *Galërkin* Neural ODEs and *Stacked* Neural ODEs.

**Gradient descent in functional space** ●  When the model parameters are depth–varying, $\theta : \mathcal{S} \rightarrow \mathbb{R}^{n_\theta}$, the nonlinear optimization problem (3) should be in principle solved by iterating a gradient descent algorithm in a functional space (Smyrlis and Zisis, 2004), e.g. $\theta_{k+1}(s) = \theta_k(s) - \eta \delta \ell_k / \delta \theta(s)$ once the Gateaux derivative $\delta \ell_k / \delta \theta(s)$ is computed. Let $\mathbb{L}_2(\mathcal{S} \rightarrow \mathbb{R}^{n_\theta})$ be the space of square–integrable functions $\mathcal{S} \rightarrow \mathbb{R}^{n_\theta}$. Hereafter, we show that if $\theta(s) \in \mathcal{W} := \mathbb{L}_2(\mathcal{S} \rightarrow \mathbb{R}^{n_\theta})$, then the loss sensitivity to $\theta(s)$ can be computed through the adjoint method.

**Theorem 1** (Infinite–Dimensional Gradients). *Consider the loss function (2) and let $\theta(s) \in \mathbb{L}_2(\mathcal{S} \rightarrow \mathbb{R}^{n_\theta})$. Then, sensitivity of $\ell$ with respect to $\theta(s)$ (i.e. directional derivative in functional space) is*

$$\frac{\delta \ell}{\delta \theta(s)} = \mathbf{a}^\top(s) \frac{\partial f_{\theta(s)}}{\partial \theta(s)} \ \text{ where } \mathbf{a}(s) \text{ satisfies } \begin{cases} \dot{\mathbf{a}}^\top(s) = -\mathbf{a}^\top(s) \frac{\partial f_{\theta(s)}}{\partial \mathbf{z}} - \frac{\partial l}{\partial \mathbf{z}} \\ \mathbf{a}^\top(S) = \frac{\partial L}{\partial \mathbf{z}(S)} \end{cases}$$

Note that, although Theorem 1 provides a constructive method to compute the loss gradient in the *infinite–dimensional* setting, its implementation requires choosing a finite dimensional approximation of the solution. We offer two alternatives: a *spectral discretization* approach relying on reformulating the problem on some functional bases and a *depth discretization* approach.

**Spectral discretization:** *Galërkin Neural ODEs* ●  The idea is to expand $\theta(s)$ on a complete orthogonal basis of a predetermined subspace of $\mathbb{L}_2(\mathcal{S} \rightarrow \mathbb{R}^{n_\theta})$ and truncate the series to the $m$-th term:

$$\theta(s) = \sum_{j=1}^{m} \alpha_j \odot \psi_j(s)$$

In this way, the problem is turned into finite dimension and the training will aim to optimize the parameters $\alpha = (\alpha_1, \ldots, \alpha_m) \in \mathbb{R}^{mn_\theta}$ whose gradients can be computed as follows

**Corollary 1** (Spectral Gradients). *Under the assumptions of Theorem 1, if $\theta(s) = \sum_{j=1}^{m} \alpha_j \odot \psi_j(s)$,*

$$\frac{\mathrm{d}\ell}{\mathrm{d}\alpha} = \int_{\mathcal{S}} \mathbf{a}^\top(\tau) \frac{\partial f_{\theta(s)}}{\partial \theta(s)} \psi(\tau) \mathrm{d}\tau, \quad \psi = (\psi_1, \ldots, \psi_m)$$

**Depth discretization:** *Stacked Neural ODEs* ●  An alternative approach to parametrize $\theta(s)$ is to assume it piecewise constant in $\mathcal{S}$, i.e. $\theta(s) = \theta_i \ \forall s \in [s_i, s_{i+1}]$ and $\mathcal{S} = \bigcup_{i=0}^{p-1}[s_i, s_{i+1}]$. It is easy to see how evaluating this model is equivalent to *stacking* $p$ Neural ODEs with constant parameters,

$$\mathbf{z}(S) = h_x(\mathbf{x}) + \sum_{i=0}^{p-1} \int_{s_1}^{s_{i+1}} f_{\theta_i}(\tau, \mathbf{x}, \mathbf{z}(\tau)) \mathrm{d}\tau$$

Here, the training is carried out optimizing the resulting $pn_\theta$ parameters using the following:

**Corollary 2** (Stacked Gradients). *Under the assumptions of Theorem 1, if $\theta(s) = \theta_i \ \forall s \in [s_i, s_{i+1}]$,*

$$\frac{\mathrm{d}\ell}{\mathrm{d}\theta_i} = -\int_{s_{i+1}}^{s_i} \mathbf{a}^\top(\tau) \frac{\partial f_{\theta_i}}{\partial \theta_i} \mathrm{d}\tau \ \text{ where } \mathbf{a}(s) \text{ satisfies } \begin{cases} \dot{\mathbf{a}}^\top(s) = -\mathbf{a}^\top(s) \frac{\partial f_{\theta_i}}{\partial \mathbf{z}} - \frac{\partial l}{\partial \mathbf{z}} & s \in [s_i, s_{i+1}] \\ \mathbf{a}^\top(S) = \frac{\partial L}{\partial \mathbf{z}(S)} \end{cases}$$

The two approaches offer different perspectives on the problem of parametrizing the evolution of $\theta(s)$; while the spectral method imposes a stronger prior to the model class, based on the chosen bases (e.g. Fourier series, Chebyshev polynomials, etc.) the depth–discretization method allows for more freedom. Further details on proofs, derivation and implementation of the two models are given in the Appendix.

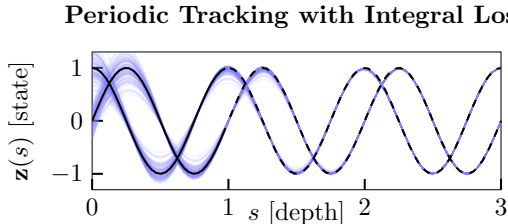

**Periodic Tracking with Integral Loss**

Figure 1: Galërkin Neural ODEs trained with integral losses accurately recover periodic signals. Blue curves correspond to different initial conditions and converge asymptotically to the reference desired trajectory.

**Tracking signals via depth–variance** Consider the problem of tracking a periodic signal $\beta(s)$. We show how this can be achieved without introducing additional inductive biases such as (Greydanus et al., 2019) through a synergistic combination of a two–layer Galërkin Neural ODEs and the generalized adjoint with integral loss $l(s) := \|\beta(s) - \mathbf{z}(s)\|_2^2$. The models, trained in $s \in [0,1]$ generalize accurately in extrapolation, recovering the dynamics. Fig.2 showcases the depth–dynamics of $\theta(s)$ for Galërkin and Stacked variants trained to solve a simple binary classification problem. Additional insights and details are reported in Appendix.

> Depth–variance brings Neural ODEs closer to the ideal continuum of neural network layers with untied weights, enhancing their expressivity.

## 4   Augmenting Neural ODEs

*Augmented Neural ODEs* (ANODEs) (Dupont et al., 2019) propose solving the *initial value problem* (IVP) in a higher dimensional space to limit the complexity of learned flows, i.e. having $n_z > n_x$. The proposed approach of the seminal paper relies on initializing to zero the $n_a := n_z - n_x$ augmented dimensions: $\mathbf{z}(0) = [\mathbf{x}, \mathbb{0}]$. We will henceforth refer to this augmentation strategy as $0$–*augmentation*. In this section we discuss alternative augmentation strategies for Neural ODEs that match or improve on $0$–augmentation in terms of performance or parameter efficiency.

**Input–layer augmentation** ● Following the standard deep learning approach of increasing layer width to achieve improved model capacity, $0$–augmentation can be generalized by introducing an

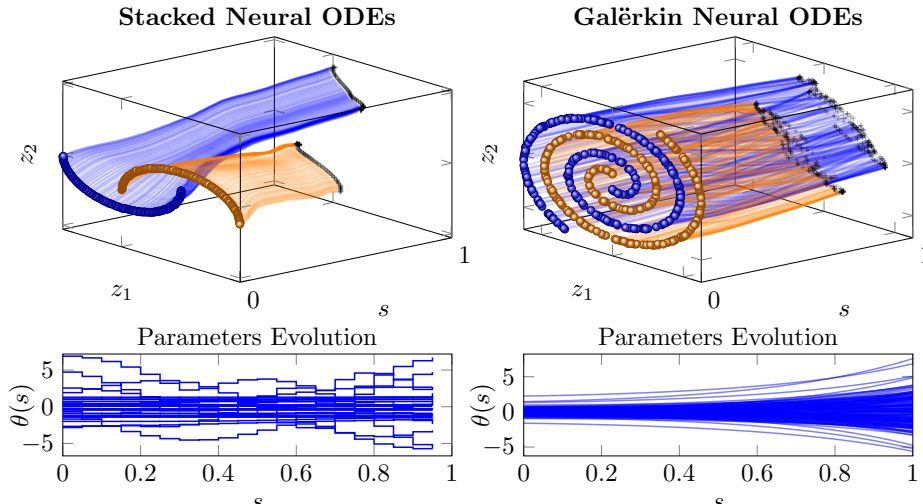

Figure 2: Galërkin and Stacked parameter-varying Neural ODE variants. Depth flows (Above) and evolution of the parameters (Below).

input network $h_x : \mathbb{R}^{n_x} \to \mathbb{R}^{n_z}$ to compute $\mathbf{z}(0)$:

$$\mathbf{z}(0) = h_x(\mathbf{x}) \tag{4}$$

leading to the general formulation of (1). This approach (4) gives the model more freedom in determining the initial condition for the IVP instead of constraining it to a concatenation of $\mathbf{x}$ and $\mathbb{0}$, at a small parameter cost if $h_x$ is, e.g., a linear layer. We refer to this type of augmentation as *input layer* (IL) augmentation and to the model as *IL–Neural ODE* (IL–NODE).

Note that 0-augmentation is compatible with the general IL formulation, as it corresponds to

$$\mathbf{x} \mapsto (\mathbf{x}, \mathbb{0}) := h_x(\mathbf{x})$$

In applications where maintaining the structure of the first $n_x$ dimensions is important, e.g. approximation of dynamical systems, a parameter efficient alternative of (4) can be obtained by modifying the input network $h_x$ to only affect the additional $n_a$ dimensions, i.e. $h_x := [\mathbf{x}, \xi(\mathbf{x})], \xi : \mathbb{R}^{n_x} \to \mathbb{R}^{n_a}$.

**Higher–order Neural ODEs** ● Further parameter efficiency can be achieved by lifting the Neural ODEs to higher orders. For example, let $\mathbf{z}(s) = [\mathbf{z}_q(s), \mathbf{z}_p(s)]$ a second–order Neural ODE of the form:

$$\ddot{\mathbf{z}}_q(s) = f_{\theta(s)}(s, \mathbf{z}(s)). \tag{5}$$

equivalent to the first–order system

$$\begin{aligned}
\dot{\mathbf{z}}_q(s) &= \mathbf{z}_p(s) \\
\dot{\mathbf{z}}_p(s) &= f_{\theta(s)}(s, \mathbf{z}_q(s), \mathbf{z}_p(s))
\end{aligned} \tag{6}$$

The above can be extended to higher–order Neural ODEs as

$$\frac{\mathrm{d}^n \mathbf{z}^1}{\mathrm{d}s^n} = f_{\theta(s)}\left(s, \mathbf{z}, \frac{\mathrm{d}\mathbf{z}^1}{\mathrm{d}s}, \cdots, \frac{\mathrm{d}^{n-1}\mathbf{z}^1}{\mathrm{d}s^{n-1}}\right), \quad \mathbf{z} = [\mathbf{z}^1, \mathbf{z}^2, \ldots, \mathbf{z}^n], \; \mathbf{z}^i \in \mathbb{R}^{n_z/n} \tag{7}$$

or, equivalently, $\dot{\mathbf{z}}^i = \mathbf{z}^{i+1}, \dot{\mathbf{z}}^n = f_{\theta(s)}(s, \mathbf{z})$. Note that the parameter efficiency of this method arises from the fact that $f_{\theta(s)} : \mathbb{R}^{n_z} \to \mathbb{R}^{n_z/n}$ instead of $\mathbb{R}^{n_z} \to \mathbb{R}^{n_z}$. A limitation of system (6) is that a naive extension to second–order requires a number of augmented dimensions $n_a = n_x$. To allow for flexible augmentations of few dimensions $n_a < n_x$, the formulation of second–order Neural ODEs can be modified to *select* only a few dimensions to have higher order dynamics. We include formulation and additional details of *selective* higher–order augmentation in the supplementary material. Finally, higher–order augmentation can itself be compatible with input–layer augmentation.

**Revisiting results for augmented Neural ODEs** In higher dimensional state spaces, such as those of image classification settings, the benefits of augmentation become subtle and manifest as performance improvements and a lower *number of function evaluations* (NFEs) (Chen et al., 2018). We revisit the image classification experiments of (Dupont et al., 2019) and evaluate four classes of depth–invariant Neural ODEs: namely, vanilla (no augmentation), ANODE (0–augmentation), IL-NODE (input–layer augmentation), and second–order. The input network $h_x$ is composed of a single, linear layer. Main objective of these experiments is to rank the efficieny of different augmentation strategies; for this reason, the setup does not involve hybrid or composite Neural ODE architectures and data augmentation.

The results for five experiments are reported in Table 4. IL–NODEs consistently preserve lower NFEs than other variants, whereas second–order Neural ODEs offer a parameter efficient alternative. The performance gap widens on CIFAR10, where the disadvantage of fixed 0 initial conditions forces 0–augmented Neural ODEs into performing a high number of function evaluations.

|  | NODE | | ANODE | | IL-NODE | | 2nd–Ord. | |
|---|---|---|---|---|---|---|---|---|
|  | MNIST | CIFAR | MNIST | CIFAR | MNIST | CIFAR | MNIST | CIFAR |
| Test Acc. | 96.8 | 58.9 | 98.9 | 70.8 | 99.1 | **73.4** | **99.2** | 72.8 |
| NFE | 98 | 93 | 71 | 169 | 44 | 65 | **43** | **59** |
| Param.[K] | 21.4 | 37.1 | 20.4 | 35.0 | 20.7 | 36.1 | **20.0** | **34.6** |

Table 1: Mean test results across 10 runs on MNIST and CIFAR. We report the mean NFE at convergence. Input layer and higher order augmentation improve task performance and preserve low NFEs at convergence.

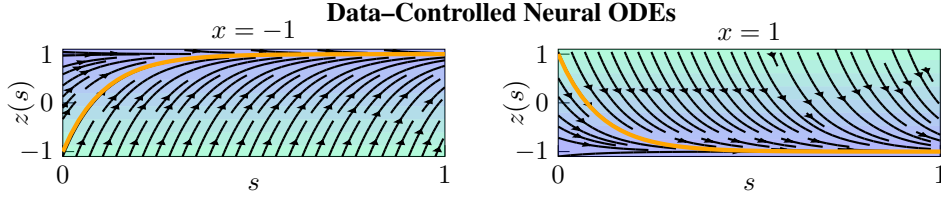

Figure 3: Depth trajectories over vector field of the *data–controlled* neural ODEs (9) for $x = 1$, $x = -1$. The model learns a family of vector fields conditioned by the input $x$ to approximate $\varphi(x)$.

It should be noted that prepending an input multi–layer neural network to the Neural ODE was the approach chosen in the experimental evaluations of the original Neural ODE paper (Chen et al., 2018) and that (Dupont et al., 2019) opted for a comparison between no input layer and 0–augmentation. However, a significant difference exists between architectures depending on the depth and expressivity of $h_x$. Indeed, utilizing non–linear and multi–layer input networks can be detrimental, as discussed in Sec. 5.

> Augmentation relieves Neural ODEs of their expressivity limitations. Learning initial conditions improves on 0–augmentation in terms of performance and NFEs.

## 5 Beyond Augmentation: Data–Control and Depth–Adaptation

Augmentation strategies are not always necessary for Neural ODEs to solve challenging tasks such as *concentric annuli* (Dupont et al., 2019). While it is indeed true that two distinct trajectories can never intersect in the state–space in the one–dimensional case, this does not necessarily hold in general. In fact, dynamics in the first two spatial dimensions are substantially different e.g no chaotic behaviors are possible (Khalil and Grizzle, 2002). In the two–dimensions of $\mathbb{R}^2$ (and so in $\mathbb{R}^n$), infinitely wider than $\mathbb{R}$, distinct trajectories of a time–varying process can well intersect in the state–space, provided that they do not pass through the same point at the same time (Khalil and Grizzle, 2002). This implies that, in turn, depth–varying models such as Galërkin Neural ODEs can solve these tasks in all dimensions but $\mathbb{R}$.

Starting from the one–dimensional case, we propose new classes of models allowing Neural ODEs to perform challenging tasks such as approximating *reflections* (Dupont et al., 2019) without the need of any augmentation.

### 5.1 Data–controlled Neural ODEs ⊙

We hereby derive a new class of models, namely *data–controlled Neural ODEs*.

To introduce the proposed approach, we start with an analytical result regarding the approximation of reflection maps such as $\varphi(x) = -x$. The proof provides a design recipe for a simple handcrafted ODE capable of approximating $\varphi$ with arbitrary accuracy by leveraging input data $x$. We denote the conditioning of the vector field with $x$ necessary to achieve the desired result as *data–control*.

This result highlights that, through data–control, Neural ODEs can arbitrarily approximate $\varphi$ without augmentation, providing a novel perspective on existing results about expressivity limitations of continuous models (Dupont et al., 2019). The result is the following:

**Proposition 2.** *For all $\epsilon > 0$, $x \in \mathbb{R}$ there exists a parameter $\theta > 0$ such that*
$$|\varphi(x) - z(1)| < \epsilon, \tag{8}$$
*where $z(1)$ is the solution of the Neural ODE*
$$\begin{cases} \dot{z}(s) = -\theta(z(s) + x) \\ z(0) = x \end{cases}, \quad s \in [0, 1] \ . \tag{9}$$

The proof is reported in the Appendix. Fig. (3) shows a version of model (9) where $\theta$ is trained with standard backpropagation. This model is indeed able to closely approximate $\varphi(x)$ without augmentation, confirming the theoretical result. From this preliminary example, we then define the

general *data–controlled* Neural ODE as

$$\dot{\mathbf{z}}(s) = f_{\theta(s)}(s, \mathbf{x}, \mathbf{z}(s))$$
$$\mathbf{z}(0) = h_x(\mathbf{x})$$
. (10)

Model (10) incorporates input data $\mathbf{x}$ into the vector field, effectively allowing the ODE to learn a *family* of vector fields instead of a single one. Direct dependence on $\mathbf{x}$ further constrains the ODE to be smooth with respect to the initial condition, acting as a regularizer. Indeed, in the experimental evaluation at the end of Sec. 5, data–controlled models recover an accurate decision boundary. Further experimental results with the latter general model on the representation of $\varphi$ are reported in the Appendix.

It should be noted that (10) does not require explicit dependence of the vector field on $\mathbf{x}$. Computationally, $\mathbf{x}$ can be passed to $f_{\theta(s)}$ in different ways, such as through an additional embedding step. In this setting, data–control offers a natural extension to conditional Neural ODEs.

**Conditional Continuous Normaling Flows**

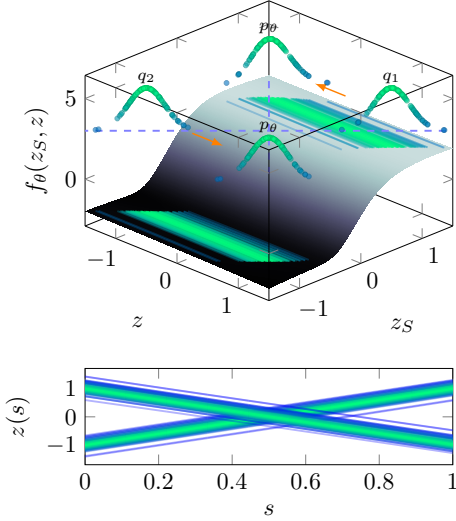

**Data–control in normalizing flows** Conditional variants of generative models can be guided to produce samples of different characteristics depending on specific requirements. Data–control can be leveraged to obtain a conditional variant of continuous normalizing flows (CNFs) (Chen et al., 2018). Here, we consider the standard setting of learning an unknown data distribution $p(\mathbf{x})$ given samples $\{\mathbf{x}_k\}_{k \in \mathcal{K}}$ through a parametrized function $p_\theta$. *Continuous normaling flows* (CNFs) (Chen et al., 2018; Grathwohl et al., 2018; Finlay et al., 2020) obtain $p_\theta$ by change of variables using the flow of an ODE to warp a (known) prior distribution $q(\mathbf{z})$, i.e. $\log p_\theta(\mathbf{x}) = \log q(\phi_S(\mathbf{x})) + \log \det |\nabla \phi_S(\mathbf{x})|$ where the $\log$ determinant of the Jacobian is computed via the fluid mechanics identity

$$\frac{\mathrm{d}}{\mathrm{d}s} \log \det |\nabla \phi_s(\mathbf{x})| = \nabla \cdot f_{\theta(t)}(s, \phi_s(\mathbf{x}))$$

, (Villani, 2003). CNFs are trained via maximum–likelihood, i.e by minimizing the Kullback–Leibler divergence between $p$ and $p_\theta$, or equivalently $\ell := -1/|\mathcal{K}| \sum_k \log p_\theta(\mathbf{x}_k)$. A CNF can be then used as generative model for

Figure 4: Data–controlled CNFs can morph prior distributions into distinct posteriors to produce conditional samples. This task often requires crossing trajectories and is not possible with vanilla CNFs.

$p_\theta(\mathbf{x})$ by sampling the known distribution $\mathbf{z}_S \sim q(\mathbf{z}_S)$ and evolve $\mathbf{z}_S$ backward in the depth domain:

$$\mathbf{z}(0) = \mathbf{z}_S + \int_S^0 f_{\theta(s)}(s, \mathbf{z}(s)) \mathrm{d}s$$

In this context, introducing data–control into $f_\theta$ allows the CNF to be conditioned with data or task information. Data–controlled CNFs can thus be used in multi–objective generative tasks e.g using a single model to sample from $N$ different distribution $p_\theta$ by warping $N$ predetermined know distributions $q_i$. We train one–dimensional, data–controlled CNFs to approximate two different data distributions $p_1$, $p_2$ by sampling from two distinct priors $q_1$, $q_2$ and conditioning the vector field with the samples $z_S$ of the prior distributions, i.e.

$$\dot{z}(s) = f_\theta(z_S, z), \quad z_S \sim q_1 \text{ or } z_S \sim q_2$$

Fig 4 shows how data–controlled CNFs are capable of conditionally sampling from two normal target data distributions. In this case we selected $p_1, p_2$ as univariate normal distributions with mean $-1$ and $1$, respectively and $q_1 \equiv p_2$, $q_2 \equiv p_1$. The resulting learned vector field strongly depends on the value of the prior sample $z_S$ and it is almost constant in $z$, meaning that the prior distributions are just shifted almost rigidly along the flow in a direction determined by the initial condition. This task is inaccessible to standard CNFs as it requires crossing flows in $z$. Indeed, the proposed benchmark represents a density estimation analogue to the crossing trajectories problem.

## 5.2 Adaptive–Depth Neural ODEs ◦

Let us come back to the approximation of $\varphi(x)$. Indeed, without incorporating input data into $f_{\theta(s)}$, it is not possible to realize a mapping $x \mapsto \phi_s(x)$ mimicking $\varphi$ due to the topology preserving property of the flows. Nevertheless, a Neural ODE can be employed to approximate $\varphi(x)$ without the need of any *crossing trajectory*. In fact, if each input is integrated for in a different depth domain, $\mathcal{S}(x) = [0, s_x^*]$, it is possible to learn $\varphi$ without crossing flows as shown in Fig. 5.

In general, we can use a hypernetwork $g$ trained to learn the integration depth of each sample. In this setting, we define the general *adaptive depth* class as Neural ODEs performing the mapping $\mathbf{x} \mapsto \phi_{g_\omega(\mathbf{x})}(\mathbf{x})$, i.e. leading to

$$\hat{\mathbf{y}} = h_y \left( h_x(\mathbf{x}) + \int_0^{g_\omega(\mathbf{x})} f_{\theta(s)}(\tau, \mathbf{x}, \mathbf{z}(\tau))d\tau \right),$$

where $g_\omega : \mathbb{R}^{n_x} \times \mathbb{R}^{n_\omega} \to \mathbb{R}$ is a neural network with trainable parameters $\omega$. Appendix B contains details on differentiation under the integral sign, required to backpropagate the loss gradients into $\omega$.

**Adaptive Integration Depth**

Inputs trajectories through network depth

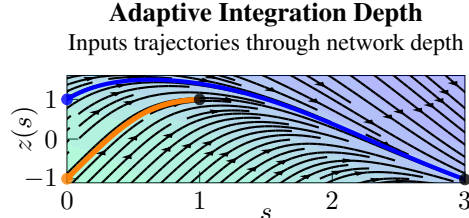

Figure 5: Depth trajectories over vector field of the *adaptive—depth* Neural ODEs. The reflection map can be learned by the proposed model. The key is to assign different integration times to the inputs, thus not requiring the intersection of trajectories.

**Experiments of non–augmented models**  We inspect the performance of different Neural ODE variants: depth–invariant, depth–variant with $s$ concatenated to $\mathbf{z}$ and passed to the vector field, Galërkin Neural ODEs and data–controlled. The concentric annuli (Dupont et al., 2019) dataset is utilized, and the models are qualitatively evaluated based on the complexity of the learned flows and on how accurately they extrapolate to unseen points, i.e. the learned decision boundaries. For Galërkin Neural ODEs, we choose a Fourier series with $m = 5$ harmonics as the eigenfunctions $\psi_k$, $k = 1, \ldots, 5$ to compute the parameters $\theta(s)$, as described in Sec. 3.

> Data–control allows Neural ODEs to learn a family of vector fields, conditioning on input data information. Depth–adaptation sidesteps known expressivity limitations of continuous–depth models.

**Original space  Flows in Latent Space**

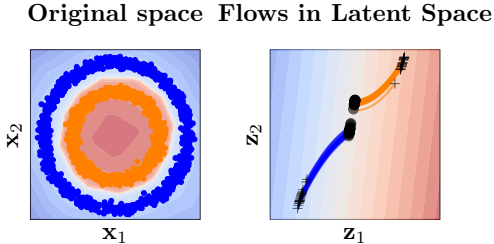

Figure 6: Solving concentric annuli without augmentation by prepending a nonlinear transformation performed by a two–layer fully–connected network.

**Mind your input networks**  An alternative approach to learning maps that prove to be challenging to approximate for vanilla Neural ODEs involves solving the ODE in a latent state space. Fig. 6 shows that with no augmentation, a network composed by a two fully–connected layers with non–linear activation followed by a Neural ODE can solve the concentric annuli problem. However, the flows learned by the Neural ODEs are superfluous: indeed, the clusters were already linearly separable after the first non–linear transformation. This example warns against superficial evaluations of Neural ODE architectures preceded or followed by several layers of non–linear input and output transformations. In these scenarios, the learned flows risk performing unnecessary transformations and in pathological cases can collapse into a simple identity map. To sidestep these issues, we propose visually inspecting trajectories or performing an ablation experiment on the Neural ODE block.

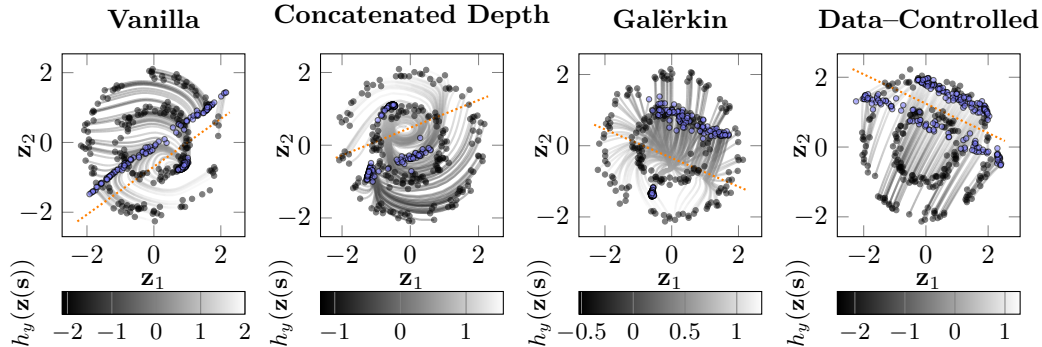

Figure 7: Depth-flows of the data in the state–space. The resulting decision boundaries of output linear layer $h_y$ are indicated by the dotted orange line.

# 6 Related Work

We include a brief history of classical approaches to dynamical system–inspired deep learning.

**A brief historical note on continuous deep learning** Continuous neural networks have a long history that goes back to continuous time variants of recurrent networks (Cohen and Grossberg, 1983). Since then, several works explored the connection between dynamical systems, control theory and machine learning (Zhang et al., 2014; Li et al., 2017; Lu et al., 2017; Weinan, 2017). (Marcus and Westervelt, 1989) provides stability analyses and introduces delays. Many of these concepts have yet to resurface in the context of Neural ODEs. Haber and Ruthotto (2017) analyzes ResNet dynamics and links stability with robustness. Injecting stability into neural networks has inspired the design of a series of architectures (Chang et al., 2019; Haber et al., 2019; Bai et al., 2019; Massaroli et al., 2020). Hauser et al. (2019) explored the algebraic structure of neural networks governed by finite difference equations, further linking discretizations of ODEs and ResNets in (Hauser et al., 2019).

Approximating ODEs with neural networks has been discussed in (Wang and Lin, 1998; Filici, 2008). (Poli et al., 2020a) explores the interplay between Neural ODEs and their solver. On the optimization front, several works leverage dynamical system formalism in continuous time (Wibisono et al., 2016; Maddison et al., 2018; Massaroli et al., 2019).

**Neural ODEs** This work concerns Neural ODEs (Chen et al., 2018) and a system–theoretic discussion of their dynamical behavior. The main focus is on Neural ODEs and not the extensions to other classes of differential equations (Li et al., 2020; Tzen and Raginsky, 2019; Jia and Benson, 2019), though the insights developed here can be broadly applied to continuous–depth models. More recently, Finlay et al. (2020) introduced regularization strategies to alleviate the heavy computational training overheads of Neural ODEs. These terms are propagated during the forward pass of the model and thus require state–augmentation. Leveraging our *generalized adjoint* formulation provides an approach to integral regularization terms without augmentation and memory overheads.

# 7 Conclusion

In this work, we establish a general system–theoretic framework for Neural ODEs and dissect it into its core components. With the aim of shining light on fundamental questions regarding depth–variance, we formulate and solve the *infinite–dimensional* problem linked to the true *deep limit* formulation of Neural ODE. We provide numerical approximations to the infinite–dimensional problem, leading to novel model variants, such as Galërkin and piecewise–constant Neural ODEs. Augmentation is developed beyond existing approaches (Dupont et al., 2019) to include *input–layer* and *higher–order* augmentation strategies showcased to be more performant and parameter efficient. Finally, the novel paradigms of data–control and depth–adaptation are introduced to perform challenging tasks such as learning *reflections* without augmentation. The code to reproduce all the experiments present in the paper is built on `TorchDyn` (Poli et al., 2020b) and `PyTorch-Lighning` (Falcon et al., 2019) libraries, can be found in the following repo: https://github.com/DiffEqML/diffeqml-research/tree/master/dissecting-neural-odes.

## Broader Impact

As continuous deep learning sees increased utilization across fields such as healthcare (Rubanova et al., 2019; Yıldız et al., 2019), it is of utmost importance that we develop appropriate tools to further our understanding of neural differential equations. The search for robustness in traditional deep learning has only recently seen a surge in ideas and proposed solutions; this work aims at providing exploratory first steps necessary to extend the discussion to this novel paradigm. The *leitmotif* of this work is injecting system–theoretic concepts into the framework of continuous models. These ideas are of foundational importance in tangential fields such control and forecasting of dynamical systems, and are routinely used to develop robust algorithms with theoretical and practical guarantees.

## Acknowledgment

This work was supported by the Basic Science Research Program through the National Research Foundation of Korea (NRF) funded by the Ministry of Education, 2018R1D1A1B07050443.

## Footnotes

[2]In practice, $s$ is often concatenated to $\mathbf{z}$ and fed to $f_\theta$.

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
