[Supplementary Material]

# Dissecting Neural ODEs
## *Supplementary Material*

## Table of Contents

## A   Proofs and Additional Theoretical Results

### A.1   Proof of Theorem 1

**Proposition 1** (Generalized Adjoint Method). *Consider the loss function* (2). *Then,*

$$\frac{\mathrm{d}\ell}{\mathrm{d}\theta} = \int_{\mathcal{S}} \mathbf{a}^\top(\tau)\frac{\partial f_\theta}{\partial \theta}\mathrm{d}\tau \; \textit{ where } \mathbf{a}(s) \textit{ satisfies } \; \begin{cases} \dot{\mathbf{a}}^\top(s) = -\mathbf{a}^\top(s)\frac{\partial f_\theta}{\partial \mathbf{z}} - \frac{\partial l}{\partial \mathbf{z}} \\ \mathbf{a}^\top(S) = \frac{\partial L}{\partial \mathbf{z}(S)} \end{cases}$$

*Proof.* Let us define a *Lagrange multiplier* or *adjoint* state $\mathbf{a}$, dual to $\mathbf{z}$. As the dual of $\mathbb{R}^{n_z}$ is $\mathbb{R}^{n_z}$ itself, $\mathbf{a} \in \mathbb{R}^{n_z}$. Moreover, let $\mathcal{L}$ be a perturbed loss function of the form

$$\mathcal{L} := \ell - \int_0^S \mathbf{a}^\top(\tau)\left[\dot{\mathbf{z}}(\tau) - f_\theta(s, \mathbf{x}_t, \mathbf{z}(\tau))\right]\mathrm{d}\tau$$

Since $\dot{\mathbf{z}} - f_\theta(s, \mathbf{x}, \mathbf{z}) = 0$ by construction, the integral term in $\mathcal{L}$ is always null and, thus, $\mathbf{a}(s)$ can be freely assigned while $\mathrm{d}\mathcal{L}/\mathrm{d}\theta = \mathrm{d}\ell/\mathrm{d}\theta$. For the sake of compactness we do not explicitly write the

dependence on variables of the considered functions unless strictly necessary. Note that,

$$\int_0^S \mathbf{a}^\top \dot{\mathbf{z}} \mathrm{d}\tau = \mathbf{a}^\top(\tau)\mathbf{z}(\tau)\big|_0^S - \int_0^S \dot{\mathbf{a}}^\top \mathbf{z} \mathrm{d}\tau$$

obtained via integration by parts. Hence,

$$
\begin{aligned}
\mathcal{L} &= \ell - \mathbf{a}^\top(\tau)\mathbf{z}(\tau)\big|_0^S + \int_0^S \left(\dot{\mathbf{a}}^\top \mathbf{z} + \mathbf{a}^\top f_\theta\right) \mathrm{d}\tau \\
&= L(\mathbf{z}(S)) - \mathbf{a}^\top(\tau)\mathbf{z}(\tau)\big|_0^S + \int_0^S \left(\dot{\mathbf{a}}^\top \mathbf{z} + \mathbf{a}^\top f_\theta + l\right) \mathrm{d}\tau
\end{aligned}
\tag{11}
$$

We can compute the gradient of $\ell$ with respect to $\theta$ as

$$
\begin{aligned}
\frac{\mathrm{d}\ell}{\mathrm{d}\theta} = \frac{\mathrm{d}\mathcal{L}}{\mathrm{d}\theta} &= \frac{\partial L(\mathbf{z}(S))}{\partial \mathbf{z}(S)}\frac{\mathrm{d}\mathbf{z}(S)}{\mathrm{d}\theta} - \mathbf{a}^\top(S)\frac{\mathrm{d}\mathbf{z}(S)}{\mathrm{d}\theta} - \mathbf{a}^\top(0)\frac{\mathrm{d}\mathbf{z}(0)}{\mathrm{d}\theta} \\
&+ \int_0^S \left[\dot{\mathbf{a}}^\top \frac{\mathrm{d}\mathbf{z}}{\mathrm{d}\theta} + \mathbf{a}^\top \left(\frac{\partial f_\theta}{\partial \theta} + \frac{\partial f_\theta}{\partial \mathbf{z}}\frac{\mathrm{d}\mathbf{z}}{\mathrm{d}\theta} + \frac{\partial f_\theta}{\partial \mathbf{x}}\frac{\mathrm{d}\mathbf{x}}{\mathrm{d}\theta} + \frac{\partial f_\theta}{\partial \tau}\frac{\mathrm{d}\tau}{\mathrm{d}\theta}\right) + \frac{\partial l}{\partial \mathbf{z}}\frac{\mathrm{d}\mathbf{z}}{\mathrm{d}\theta} + \frac{\partial l}{\partial \tau}\frac{\mathrm{d}\tau}{\mathrm{d}\theta}\right] \mathrm{d}\tau
\end{aligned}
$$

which, by reorganizing the terms, yields to

$$
\begin{aligned}
\frac{\mathrm{d}\ell}{\mathrm{d}\theta} = &\left[\frac{\partial L}{\partial \mathbf{z}(S)} - \mathbf{a}^\top(S)\right]\frac{\mathrm{d}\mathbf{z}(S)}{\mathrm{d}\theta} + \\
&+ \int_0^S \left(\dot{\mathbf{a}}^\top + \mathbf{a}^\top \frac{\partial f_\theta}{\partial \mathbf{z}} + \frac{\partial l}{\partial \mathbf{z}}\right)\frac{\mathrm{d}\mathbf{z}}{\mathrm{d}\theta}\mathrm{d}\tau \\
&+ \int_0^S \mathbf{a}^\top \frac{\partial f_\theta}{\partial \theta}\mathrm{d}\tau
\end{aligned}
\tag{12}
$$

Now, if $\mathbf{a}(s)$ satisfies the *final* value problem

$$\dot{\mathbf{a}}^\top(s) = -\mathbf{a}^\top(s)\frac{\partial f_\theta}{\partial \mathbf{z}} - \frac{\partial l}{\partial \mathbf{z}}, \quad \mathbf{a}^\top(S) = \frac{\partial L}{\partial \mathbf{z}(S)} \tag{13}$$

to be solved backward in $[0, S]$; then (12) reduces to

$$\frac{\mathrm{d}\ell}{\mathrm{d}\theta} = \int_0^S \mathbf{a}^\top \frac{\partial f_\theta}{\partial \theta}\mathrm{d}\tau \tag{14}$$

proving the result. □

**Remark 1** (Implementation of the generalized adjoint method). *Note that, similarly to Chen et al. (2018), the gradient (14) is practically computed by defining the parameters adjoint state $\mathbf{a}_\theta$ and solving backward the system of ODEs*

$$
\begin{aligned}
\dot{\mathbf{a}}^\top &= -\mathbf{a}^\top \frac{\partial f_\theta}{\partial \mathbf{z}} - \frac{\partial l}{\partial \mathbf{z}}, \quad \mathbf{a}^\top(S) = \frac{\partial L}{\partial \mathbf{z}(S)} \\
\dot{\mathbf{a}}_\theta^\top &= -\mathbf{a}^\top \frac{\partial f_\theta}{\partial \theta}, \qquad\quad \mathbf{a}_\theta(S) = \mathbb{0}_{n_\theta}
\end{aligned}
\tag{15}
$$

*Then,*

$$\frac{\mathrm{d}\ell}{\mathrm{d}\theta} = \mathbf{a}_\theta(0).$$

## A.2   Proof of Theorem 1

**Theorem 1** (Infinite–Dimensional Gradients). *Consider the loss function (2) and let $\theta(s) \in \mathbb{L}_2(\mathcal{S} \to \mathbb{R}^{n_\theta})$. Then, sensitivity of $\ell$ with respect to $\theta(s)$ (i.e. directional derivative in functional space) is*

$$\frac{\delta\ell}{\delta\theta(s)} = \mathbf{a}^\top(s)\frac{\partial f_{\theta(s)}}{\partial \theta(s)} \text{ where } \mathbf{a}(s) \text{ satisfies} \begin{cases} \dot{\mathbf{a}}^\top(s) = -\mathbf{a}^\top(s)\frac{\partial f_{\theta(s)}}{\partial \mathbf{z}} - \frac{\partial l}{\partial \mathbf{z}} \\ \mathbf{a}^\top(S) = \frac{\partial L}{\partial \mathbf{z}(S)} \end{cases}$$

*Proof.* The proof follows the same steps of the one of Theorem 1 up to (11). However, here $\theta(s) \in \mathbb{L}_2$ and the loss sensitivity to $\theta(s)$ corresponds to the directional (Gateaux) derivative $\delta\ell/\delta\theta(s)$ in $\mathbb{L}_2$

derived as follows. We start by computing the total variation of $\ell$:

$$\delta\ell = \frac{\partial L}{\partial \mathbf{z}(S)}\delta\mathbf{z}(S) - \mathbf{a}^\top(s)(\delta\mathbf{z}(S) - \delta\mathbf{z}(0))$$

$$+ \int_0^S \left[\dot{\mathbf{a}}^\top(\tau)\delta\mathbf{z}(\tau) + \mathbf{a}^\top(\tau)\left(\frac{\partial f_{\theta(\tau)}}{\partial \mathbf{z}(\tau)}\delta\mathbf{z}(\tau) + \frac{\partial f_{\theta(\tau)}}{\partial \theta(\tau)}\delta\theta(\tau)\right) + \frac{\partial l}{\partial \mathbf{z}(\tau)}\delta\mathbf{z}(\tau)\right]\mathrm{d}\tau$$

Thus,

$$\frac{\delta\ell}{\delta\theta(s)} = \left[\frac{\partial L}{\partial \mathbf{z}(S)} - \mathbf{a}^\top(s)\right]\frac{\delta\mathbf{z}(S)}{\delta\theta(s)} + \frac{\delta\mathbf{z}(0)}{\delta\theta(s)}$$

$$+ \int_0^S \left[\dot{\mathbf{a}}^\top(\tau)\frac{\delta\mathbf{z}(\tau)}{\delta\theta(s)} + \mathbf{a}^\top(\tau)\left(\frac{\partial f_{\theta(\tau)}}{\partial \mathbf{z}(\tau)}\frac{\delta\mathbf{z}(\tau)}{\delta\theta(s)} + \frac{\partial f_{\theta(\tau)}}{\partial \theta(\tau)}\frac{\delta\theta(\tau)}{\delta\theta(s)}\right) + \frac{\partial l}{\partial \mathbf{z}(\tau)}\frac{\delta\mathbf{z}(\tau)}{\delta\theta(s)}\right]\mathrm{d}\tau$$

Since it must hold

$$\int \frac{\delta\theta(\tau)}{\delta\theta(s)}\mathrm{d}\tau = 1,$$

then, model class choice $\theta(s) \in \mathbb{L}_2$ implies

$$\frac{\delta\theta(\tau)}{\delta\theta(s)} = \delta(\tau - s)$$

where $\delta(\tau - s)$ is the Dirac's delta. Therefore, it holds

$$\frac{\delta\ell}{\delta\theta(s)} = \left[\frac{\partial L}{\partial \mathbf{z}(S)} - \mathbf{a}^\top(s)\right]\frac{\delta\mathbf{z}(S)}{\delta\theta(s)} + \frac{\delta\mathbf{z}(0)}{\delta\theta(s)}$$

$$+ \int_0^S \left[\dot{\mathbf{a}}^\top(\tau)\frac{\delta\mathbf{z}(\tau)}{\delta\theta(s)} + \mathbf{a}^\top(\tau)\left(\frac{\partial f_{\theta(\tau)}}{\partial \mathbf{z}(\tau)}\frac{\delta\mathbf{z}(\tau)}{\delta\theta(s)} + \frac{\partial f_{\theta(\tau)}}{\partial \theta(\tau)}\delta(\tau - s)\right) + \frac{\partial l}{\partial \mathbf{z}(\tau)}\frac{\delta\mathbf{z}(\tau)}{\delta\theta(s)}\right]\mathrm{d}\tau$$

and, finally

$$\frac{\delta\ell}{\delta\theta(s)} = \left[\frac{\partial L}{\partial \mathbf{z}(S)} - \mathbf{a}^\top(s)\right]\frac{\delta\mathbf{z}(S)}{\delta\theta(s)} + \frac{\delta\mathbf{z}(0)}{\delta\theta(s)}$$

$$+ \int_0^S \left(\dot{\mathbf{a}}^\top(\tau) + \mathbf{a}^\top(\tau)\frac{\partial f_{\theta(\tau)}}{\partial \mathbf{z}(\tau)} + \frac{\partial l}{\partial \mathbf{z}(\tau)}\right)\frac{\delta\mathbf{z}(\tau)}{\delta\theta(s)}\mathrm{d}\tau$$

$$+ \mathbf{a}^\top(s)\frac{\partial f_{\theta(s)}}{\partial \theta(s)}$$

Hence, if for any $s \in \mathcal{S}$ the adjoint state $\mathbf{a}(s)$ satisfies

$$\dot{\mathbf{a}}^\top = -\mathbf{a}^\top \frac{\partial f_{\theta(s)}}{\partial \mathbf{z}} - \frac{\partial l}{\partial \mathbf{z}}, \quad \mathbf{a}^\top(S) = \frac{\partial L}{\partial \mathbf{z}(S)}$$

we have

$$\frac{\delta\ell}{\delta\theta(s)} = \mathbf{a}^\top(s)\frac{\partial f_{\theta(s)}}{\partial \theta(s)}$$

$\square$

## A.3 Proof of Corollary 1

**Corollary 1** (Spectral Gradients). *Under the assumptions of Theorem 1, if $\theta(s) = \sum_{j=1}^m \alpha_j \odot \psi_j(s)$,*

$$\frac{\mathrm{d}\ell}{\mathrm{d}\alpha} = \int_{\mathcal{S}} \mathbf{a}^\top(\tau)\frac{\partial f_{\theta(s)}}{\partial \theta(s)}\psi(\tau)\mathrm{d}\tau, \quad \psi = (\psi_1, \ldots, \psi_m)$$

*Proof.* The proof follows naturally from Theorem 1 by noticing that if $\theta(s)$ has some parametrization $\theta = \theta(s, \mu)$ with parameters $\mu \in \mathbb{R}^{n_\mu}$, then,

$$\frac{\mathrm{d}\ell}{\mathrm{d}\mu} = \int_0^S \mathbf{a}^\top(\tau)\frac{\partial f_\theta}{\partial \theta}\frac{\partial \theta}{\partial \mu}\mathrm{d}\tau \tag{16}$$

Therefore, if

$$\theta(s) = \sum_{j=1}^m \alpha_j \odot \psi_j(s),$$

the loss gradient with respect to the parameters $\alpha := (\alpha_1, \ldots, \alpha_m) \in \mathbb{R}^{mn_\theta}$ is computed as

$$\frac{\mathrm{d}\ell}{\mathrm{d}\alpha} = \int_0^S \mathbf{a}^\top(\tau) \frac{\partial f_{\theta(\tau)}}{\partial \theta(\tau)} \frac{\partial \theta(s)}{\partial \alpha} \mathrm{d}\tau$$

$$= \int_0^S \mathbf{a}^\top(\tau) \frac{\partial f_{\theta(\tau)}}{\partial \theta(\tau)} \psi \mathrm{d}\tau$$

being $\psi := (\psi_1, \ldots, \psi_m)$. □

**Remark 2** (Choose your parametrization). *A further insight from this result, which paves the way to future developments, is that we can easily compute the loss gradients with respect to any parametrization of $\theta(s)$ through (16)*

## A.4 Proof of Corollary 2

**Corollary 2** (Stacked Gradients). *Under the assumptions of Theorem 1, if $\theta(s) = \theta_i \ \forall s \in [s_i, s_{i+1}]$,*

$$\frac{\mathrm{d}\ell}{\mathrm{d}\theta_i} = -\int_{s_{i+1}}^{s_i} \mathbf{a}^\top(\tau) \frac{\partial f_{\theta_i}}{\partial \theta_i} \mathrm{d}\tau \ \text{ where } \mathbf{a}(s) \text{ satisfies } \begin{cases} \dot{\mathbf{a}}^\top(s) = -\mathbf{a}^\top(s)\frac{\partial f_{\theta_i}}{\partial \mathbf{z}} - \frac{\partial l}{\partial \mathbf{z}} & s \in [s_i, s_{i+1}] \\ \mathbf{a}^\top(S) = \frac{\partial L}{\partial \mathbf{z}(S)} \end{cases}$$

*Proof.* The proof follows from the one of Theorems 1 and 1 by recalling the solution of the stacked neural ODEs:

$$\mathbf{z}(S) = h_x(\mathbf{x}) + \sum_{i=0}^{p-1} \int_{s_1}^{s_{i+1}} f_{\theta_i}(\tau, \mathbf{x}, \mathbf{z}(\tau)) \mathrm{d}\tau$$

We can recover a relation similar to (12)

$$\frac{\mathrm{d}\ell}{\mathrm{d}\theta_i} = \left[ \frac{\partial L}{\partial \mathbf{z}(S)} - \mathbf{a}^\top(S) \right] \frac{\mathrm{d}\mathbf{z}(S)}{\mathrm{d}\theta_i} +$$

$$+ \sum_{j=0}^{p-1} \int_{s_j}^{s_{j+1}} \left( \dot{\mathbf{a}}^\top + \mathbf{a}^\top \frac{\partial f_{\theta_j}}{\partial \mathbf{z}} + \frac{\partial l}{\partial \mathbf{z}} \right) \frac{\mathrm{d}\mathbf{z}}{\mathrm{d}\theta_i} \mathrm{d}\tau$$

$$+ \sum_{j=0}^{p-1} \int_{s_j}^{s_{j+1}} \mathbf{a}^\top \frac{\partial f_{\theta_j}}{\partial \theta_i} \mathrm{d}\tau$$

Since

$$\forall j = 0, \ldots, p-1 \quad \frac{\partial f_{\theta_j}}{\partial \theta_i} \neq \mathbb{0} \Leftrightarrow j = i,$$

we have

$$\sum_{j=0}^{p-1} \int_{s_j}^{s_{j+1}} \mathbf{a}^\top \frac{\partial f_{\theta_j}}{\partial \theta_i} \mathrm{d}\tau = \int_{s_i}^{s_{i+1}} \mathbf{a}^\top \frac{\partial f_{\theta_i}}{\partial \theta_i} \mathrm{d}\tau = -\int_{s_{i+1}}^{s_i} \mathbf{a}^\top \frac{\partial f_{\theta_i}}{\partial \theta_i} \mathrm{d}\tau$$

which leads to the result by assuming $\mathbf{a}(\tau)$ to satisfy

$$\dot{\mathbf{a}}^\top(s) = -\mathbf{a}^\top(s) \frac{\partial f_{\theta_i}}{\partial \mathbf{z}} - \frac{\partial l}{\partial \mathbf{z}} \quad s \in [s_i, s_{i+1}]$$

$$\mathbf{a}^\top(S) = \frac{\partial L}{\partial \mathbf{z}(S)}$$

□

## A.5 Proof of Theorem 2

**Proposition 2.** *For all $\epsilon > 0$, $x \in \mathbb{R}$ there exists a parameter $\theta > 0$ such that*

$$|\varphi(x) - z(1)| < \epsilon, \tag{8}$$

*where $z(1)$ is the solution of the Neural ODE*

$$\begin{cases} \dot{z}(s) = -\theta(z(s) + x) \\ z(0) = x \end{cases}, \quad s \in [0, 1] . \tag{9}$$

*Proof.* The general solution of (9) is

$$z(s) = x(2e^{-\theta s} - 1)$$

Thus,

$$e = \varphi(x) - z(1) = x + x(2e^{-\theta} - 1) = 2xe^{-\theta}$$
$$\Leftrightarrow |e| = 2|x|e^{-\theta}$$

It follows that

$$2|x|e^{-\theta} < \epsilon$$
$$\Leftrightarrow e^{-\theta} < \frac{\epsilon}{2|x|}$$
$$\Leftrightarrow \theta > -\ln\left(\frac{\epsilon}{2|x|}\right)$$

$\square$

## A.6 Additional Theoretical Results

### A.6.1 Explicit Parameter Dependence of the Loss

Note that, in both the seminal paper from Chen et al. (2018) and Theorem 1 the loss function was consider without explicit dependence on the parameters. However, in practical applications (see, e.g. (Finlay et al., 2020)) the loss has this explicit dependence:

$$\ell = L(\mathbf{z}(S), \theta) + \int_{\mathcal{S}} l(s, \mathbf{z}(\tau), \theta) d\tau, \tag{17}$$

In this case we need to modify the adjoint gradients accordingly

**Theorem 2** (Generalized Adjoint Method with Parameter–Dependent Loss). *Consider the loss function* (17). *Then,*

$$\frac{d\ell}{d\theta} = \frac{\partial L}{\partial \theta} + \int_{\mathcal{S}} \left( \mathbf{a}^\top(\tau) \frac{\partial f_\theta}{\partial \theta} + \frac{\partial l}{\partial \theta} \right) d\tau$$

*where* $\mathbf{a}(s)$ *satifies* (13).

*Proof.* The proof follows immediately from Theorem 1 by noticing that, with the explicit dependence on $\theta$ of $\ell$, (12) would become

$$\frac{d\ell}{d\theta} = \frac{\partial L}{\partial \theta}$$
$$+ \left[ \frac{\partial L}{\partial \mathbf{z}(S)} - \mathbf{a}^\top(S) \right] \frac{d\mathbf{z}(S)}{d\theta} +$$
$$+ \int_0^S \left( \dot{\mathbf{a}}^\top + \mathbf{a}^\top \frac{\partial f_\theta}{\partial \mathbf{z}} + \frac{\partial l}{\partial \mathbf{z}} \right) \frac{d\mathbf{z}}{d\theta} d\tau$$
$$+ \int_0^S \left( \mathbf{a}^\top \frac{\partial f_\theta}{\partial \theta} + \frac{\partial l}{\partial \theta} \right) d\tau$$

leading to the result. $\square$

In the depth–variant case where we might consider a loss function of type

$$\ell = L(\mathbf{z}(S), \theta(S)) + \int_{\mathcal{S}} l(\mathbf{z}(\tau), \theta(\tau)) d\tau \tag{18}$$

a similar result can be obtained for the infinite–dimensional adjoint.

### A.6.2 Integration Bound Gradients

It is also possible to obtain the loss gradient with respect to the integration bound $S$.

**Theorem 3** (Integration Bound Gradient). *Consider a loss function* 2. *Then,*

$$\frac{d\ell}{dS} = \frac{\partial L}{\partial \mathbf{z}(S)} f_{\theta(S)}(S, \mathbf{x}, \mathbf{z}(S)) + l(\mathbf{z}(S))$$

*Proof.*

$$\frac{\mathrm{d}\ell}{\mathrm{d}S} = \frac{\partial L}{\partial \mathbf{z}(S)}\frac{\mathrm{d}\mathbf{z}(S)}{\mathrm{d}S} + \frac{\mathrm{d}}{\mathrm{d}S}\int_0^S l(\mathbf{z}(\tau))\mathrm{d}\tau$$

$$= \frac{\partial L}{\partial \mathbf{z}(S)}\frac{\mathrm{d}}{\mathrm{d}S}\left(h_x(\mathbf{x}) + \int_0^S f_{\theta(\tau)}(\tau, \mathbf{x}, \mathbf{z}(\tau))\right) + \frac{\mathrm{d}}{\mathrm{d}S}\int_0^S l(\mathbf{z}(\tau))\mathrm{d}\tau$$

Therefore, by applying the Leibniz integral rule we obtain

$$\frac{\mathrm{d}\ell}{\mathrm{d}S} = \frac{\partial L}{\partial \mathbf{z}(S)}f_{\theta(S)}(S, \mathbf{x}, \mathbf{z}(S)) + l(\mathbf{z}(S))$$

$\square$

# B  Practical Insights for Neural ODEs

## B.1  Augmentation

**Augmenting convolution and graph based architectures**  In the case of *convolutional neural network* (CNN) or *graph neural network* (GNN) architectures, augmentation can be performed along different dimensions i.e. *channel*, heigth, width or similarly node features or number of nodes. The most physically consistent approach, employed in (Dupont et al., 2019) for CNNs, is augmenting along the *channel* dimension, equivalent to providing each pixel in the image additional states. By viewing an image as a lattice graph, the generalization to GNN–based Neural ODEs (Poli et al., 2019) operating on arbitrary graphs can be achieved by augmenting each node feature with $n_a$ additional states.

**Selective higher–order**  A limitation of system (6) is that a naive extension to second–order requires a number of augmented dimensions $n_a = n_z/2$. To allow for flexible augmentations of few dimensions $n_a < n_z/2$, the formulation of second–order Neural ODEs can be modified as follows. Let $\mathbf{z} := (\mathbf{z}_q, \mathbf{z}_p, \bar{\mathbf{z}})$, $\mathbf{z}_q, \mathbf{z}_p \in \mathbb{R}^{n_a/2}$, $\bar{\mathbf{z}} \in \mathbb{R}^{n_z - n_a}$. We can decide to give second order dynamics only to the first $n_a$ states while the dynamics of other $n_z - n_a$ states is free. Therefore, this approach yields

$$\begin{bmatrix} \dot{\mathbf{z}}_q \\ \dot{\mathbf{z}}_p \\ \dot{\bar{\mathbf{z}}} \end{bmatrix} = \begin{bmatrix} \mathbf{z}_p \\ f_{\theta(s)}^p(s, \mathbf{z}) \\ \bar{f}_{\theta(s)}(s, \mathbf{z}) \end{bmatrix}, \tag{19}$$

A similar argument could be applied to orders higher than two. *Selective higher–order* Neural ODEs are compatible with input layer augmentation.

## B.2  Activations

**Mind your activation**  We investigate the effects of appending an activation function to the last layer of $f_\theta$. The chosen nonlinearity will strongly affect the "shape" of the vector field and, as a

Figure 8: Depth trajectories of the hidden state and relative vector fields $f_\theta(\mathbf{z})$ for different activation functions in a nonlinear classification task. It can be noticed how the models with tanh and ELU outperform the others, as $f_\theta$ is able to steer $\mathbf{z}$ along negative directions.

consequence, the flows learnable by the model. Therefore, while designing $f_\theta$ as a multi–layer neural network, it is generally advisable to append a linear layer to maximize the expressiveness of the underlying vector field. In some applications, conditioning the vector field (and thus the flows) with a specific nonlinearities can be desirable, e.g., when there exist priors on the desired transformation, such as boundedness of the vector field.

**Effects of activations** In order to compare the effect of different activation functions in the last layer of $f_\theta$, we set up a nonlinear classification task with the half–moons dataset. For the sake of completeness, we selected activations of different types, i.e., The dataset is comprised of $2^{13}$ data

| Activation | Type |
|---|---|
| Hyperbolic tangent (**tanh**) | bounded |
| **Sigmoid** | bounded, non–negative output |
| **ReLU** | unbounded, non–negative output |
| **Softplus** | unbounded, non–negative output |
| **ELU** | lower–bounded |

points. We utilize the entire dataset for training and evaluation since the experiment has the aim of delivering a qualitative description of the learned vector fields. $f_\theta$ has been selected as a multilayer perceptron with two hidden layers of 16 neurons each. The training has been carried out using Adam (Kingma and Ba, 2014) optimizer with learning rate $10^{-3}$ and weight decay set to $10^{-4}$.

In Figure 8 we see how different activation functions in the last layer of $f_\theta$ condition the vector fields and the depth evolution of the hidden state in the classification of nonlinearly separable data. It is worth to be noticed that the models with better performance are the ones with *hyperbolic tangent* (tanh) and ELU (Clevert et al., 2015) as the vector field can assume both positive and negative values and, thus, can "force" the hidden state in different directions. On the other hand, with sigmoid, ReLU or softplus (Zheng et al., 2015), the vector field is nonnegative in all directions and thus has limited freedom. Further, Figure 9 shows how different activation functions shape the vector field and as a result the decision boundary.

## B.3 Regularization for Stability

The concept of stability can be used to regularize Neural ODEs through a variety of additional terms or different formulations (Finlay et al., 2020; Massaroli et al., 2020). (Finlay et al., 2020) proposes minimizing a loss term:

$$\ell_{\texttt{reg}} = \int_\mathcal{S} \|f_{\theta(\tau)}(\tau, \mathbf{x}, \mathbf{z}(\tau)\|_2 \mathrm{d}\tau, \tag{20}$$

to achieve stability. A simple alternative stabilizing regularization term can be considered at no significant additional computational cost:

$$\ell_{\texttt{reg}} = \big\|f_{\theta(S)}(S, \mathbf{x}, \mathbf{z}(S))\big\|_2, \tag{21}$$

which penalizes non–convergence to some fixed point of $f_\theta$ at $s = S$. The above can also be seen as a cheaper alternative to the kinetic energy regularization proposed in (Finlay et al., 2020).

## B.4 Approximation Capabilities

Vanilla Neural ODEs are not, in general, universal function approximators (UFAs) (Zhang et al., 2019a). Besides some recent works on the topic (Zhang et al., 2019a; Li et al., 2019) this apparent limitation is still not well–understood in the context of continuous–depth models. When Neural ODEs are employed as general–purpose black–box modules, some assurances on the approximation capabilities of the model are necessary. Let $n_z := n_x + 1$ and let $\mathbf{z} := (\mathbf{z}_x, z_a)$ $(\mathbf{z}_x \in \mathbb{R}^{n_x}, \ z_a \in \mathbb{R})$. (Zhang et al., 2019a) noticed that a depth–invariant augmented Neural ODE

$$\begin{bmatrix} \dot{\mathbf{z}}_x \\ \dot{z}_a \end{bmatrix} = \begin{bmatrix} \mathbb{0}_{n_x} \\ f_\theta(\mathbf{z}_x) \end{bmatrix}, \quad \begin{bmatrix} \mathbf{z}_x(0) \\ z_a(0) \end{bmatrix} = \begin{bmatrix} \mathbf{x} \\ 0 \end{bmatrix}, \ \ s \in [0, 1] \tag{22}$$

where the output is picked as $\hat{y} := z_a(1)$, can approximate any function $\Psi : \mathbb{R}^{n_x} \to \mathbb{R}$ provided that the neural network $f_\theta(\mathbf{x})$ is an approximator of $\Psi$, since $z_a(1) = f_\theta(\mathbf{x})$, mimicking the mapping

$\mathbf{x} \mapsto f_\theta(\mathbf{x})$. Although this simple result is not sufficient to provide a constructive blueprint to the design of Neural ODE models, it suggests the following (open) questions:

- Why should we use a Neural ODE if its vector field can solve the approximation problem as a standalone neural network?

- Can Neural ODEs be UFAs with non-UFA vector fields?

On the other hand, if Neural ODEs are used for model discovery or observation of dynamical systems, requiring an UFA neural network to parametrize the model provides it with the ability to approximate arbitrary dynamical systems.

### B.5 Example Implementation of Data–Control

We report here a short PyTorch code snippet detailing the implementation of the simplest *data–controlled Neural ODE* variant, accompanied, for further accessibility, by a brief text description.

```python
class DC_DEFunc(nn.Module):
    """PyTorch implementation of data--controlled $f_\theta$"""
    def __init__(self, f):
        super().__init__()
        self.f = f

    def forward(self, s, z):
    """Forward is called by the ODE solver repeatedly"""
        self.nfe += 1
        # data-control step:
        # alternatives include embeddings of input data `x` i.e g(x)
        # or addition `x + z`
        z = torch.cat([z, self.x], 1)
        dz = self.f(z)
        return dz
```

where the initial condition $\mathbf{x}$ is passed to the model at the start of the integration at $s = 0$. The information contained is thus passed repeatedly to the function $f_\theta$, conditioning the dynamics. It should be noted that even in the case of concatenation of $\mathbf{x}$ and $\mathbf{z}(s)$, the above is not a form of augmentation, since the state itself is not given additional dimensions during forward propagation. In fact, the dynamics take the form of a function $f_\theta : \mathbb{R}^{n_x} \times \mathbb{R}^{n_z} \to \mathbb{R}^{n_z}$ instead of $f_\theta : \mathbb{R}^{n_a} \times \mathbb{R}^{n_x} \to \mathbb{R}^{n_a} \times \mathbb{R}^{n_x}$ as is the case for general first–order augmentation with $n_z = n_x + n_a$.

## C  Experimental Details

**Computational resources**   The experiments were carried out on a cluster of two NVIDIA® TITAN RTX GPUs with CUDA 10.1 and INTEL® I9 10980XE CPU. All Neural ODEs were trained on GPU. The code was built upon Pytorch's `torchdyn` library for neural differential equations (Poli et al., 2020b).

**General experimental setup**   We report here general information about the experiments. All Neural ODEs are solved numerically via the *Dormand–Prince* method (Prince and Dormand, 1981). We refer to *concat* as the depth–variant Neural ODE variants where the depth–variable $s$ is concatenated to $\mathbf{z}(s)$ as done in (Chen et al., 2018). Furthermore, we denote Galërkin Neural ODEs as *GalNODE* for convenience.

**Benchmark problems**   Throughout the paper we extensively utilize the *concentric annuli* benchmark task introduced in (Dupont et al., 2019) is used extensively. Namely, given $r > 0$ define $\varphi : \mathbb{R}^n \to \mathbb{Z}$

$$\varphi(\mathbf{x}) = \begin{cases} -1 & \|\mathbf{x}\|_2 < r \\ 1 & \|\mathbf{x}\|_2 \geq r \end{cases} . \tag{23}$$

Figure 9: Decision boundaries learned by the vector field of a Neural ODE are directly conditioned by the choice of activation function.

We consider learning the map $\varphi(\mathbf{x})$ with Neural ODEs prepending a linear layer $\mathbb{R}^n \to \mathbb{R}$. Notice that $\varphi$ has been slightly modified with respect to (Dupont et al., 2019), to be *well–defined* in its domain. For the one–dimensional case, we will often instead refer to the map $\varphi(x) = -x$ as the *crossing trajectories* problem. The optimization is carried out by minimizing *mean squared error* (MSE) losses of model outputs and mapping $\varphi$.

### C.1 Experiments of Section 3

**Trajectory tracking** Consider the problem of tracking a periodic signal $\beta(s)$. We show how this can be achieved without introducing additional inductive biases such as (Greydanus et al., 2019) through a synergistic combination of a two–layer Galërkin Neural ODEs and the generalized adjoint with integral loss $l(s) := \|\beta(s) - \mathbf{z}(s)\|_2^2$. In particular, we construct a two–layer Galërkin Neural ODE with Fourier series and $m = 2$ harmonics as the eigenfunctions. The training is carried out for 1000 epochs with learning rate $10^{-3}$. The practical implementation of the generalized adjoint necessary to distribute the loss across the depth domain is discussed in Appendix A.

The models, trained in $s \in [0, 1]$ generalize accurately when tasked to perform long trajectory extrapolation of several seconds.

**Depth–varying classification** We showcase how different discretization options of the functional optimization problem discussed in Sec. 3 affect the final dynamics of $\theta(s)$. Namely, we consider a simple binary classification on the *nested spirals* problem, training all models for 300 epochs and learning rate $5 \cdot 10^{-3}$. Galërking Neural ODEs are equipped with a polynomial basis with $m = 10$. The Fig.s in Sec 3 reveal the different nature of $\theta(s)$ depending on model choice: depth–discretization of *Stacked* yields a flexible, though lower resolution form of $\theta(s)$, whereas spectral discretizations limit the functional form of $\theta(s)$ to the span of a chosen eigenbasis.

**Mind your input network experiments** We tackle the *concentric annuli* task with a Neural ODE preceded by a simple two–layer neural network with 16 units and ReLU activation. The second layer is linear.

### C.2 Experiments of Section 4

**Image classification** We use *AdamW* with learning rate $10^{-3}$, batch size 64, weight decay $5 * 10^{-4}$ and a learning rate step schedule with multiplicative factor $\gamma = 0.9$ every 5 epochs. We train each model for 20 epochs. The vector fields $f_\theta$ are parametrized by 3–layer depth–invariant CNNs, with each layer followed by an instance normalization layer. The choice of depth–invariance is motivated by the discussion carried out in Section 5: both augmentation and depth–variance can relieve approximation limitations of vanilla, depth–invariant Neural ODEs. As a result, including both renders the ablation study for augmentation strategies less accurate. We note that the results of this ablation analysis do not utilize any form of data augmentation; data augmentation can indeed be introduced to further improve performance.

For input layer augmented Neural ODE models, namely IL–NODE and 2nd order, we prepend to the Neural ODE a single, linear CNN layer. In the case of 2nd order models, we use input layer augmentation for the positions and initialize the velocities at 0. The hidden channel dimension of the CNN parametrizing $f_\theta$ in augmented models is set to 32 on MNIST and 42 on CIFAR; vanilla Neural ODEs, on the other hand, are equipped with dimensions 42 and 62 for a fair comparison. The output class probabilities are then computed by mapping the output of the Neural ODE through average pooling followed by a linear layer. Second order Neural ODEs, *2nd*, use $f_\theta$ to compute the vector

Figure 10: Depth evolution over the learned vector fields of the standard models: depth–invariant and depth–variant ("concat" $f_\theta(s, z(s))$ and GalNODE $f_\theta(s, z(s))$). As expected the Neural ODE cannot approximate the map $\varphi(x) = -x$.

field of velocities: therefore, the output of $f_\theta$ is $n_x/2$–dimensional, and the remaining $n_x/2$ outputs to concatenate (vector field of *positions*) are obtained as the last $n_x/2$ elements of **z**.

We note that vanilla Neural ODEs are capable of convergence without any spikes in loss or NFEs. We speculate the numerical issues encountered in (Dupont et al., 2019) to be a consequence of the specific neural network architecture used to parametrize the vector field $f_\theta$, which employed an excessive number of channels inside $f_\theta$, i.e 92.

### C.3 Experiments of Section 5

**Experiments on crossing trajectories**   We trained both current state–of–the–art as well as proposed models to learn the map $\varphi(x) = -x$. We created a training dataset sampling $x$ equally spaced in $[-1, 1]$. The models have been trained to minimize L1 losses using Adam (Kingma and Ba, 2014) with learning rate $lr = 10^{-3}$ and weight decay $10^{-5}$ for 1000 epochs using the whole batch. We trained vanilla Neural ODEs, i.e. both depth–invariant and depth variant models ("concat" and GalNODE). As expected, these models cannot approximate $\varphi$. Both depth–invariant and *concat* have been selected with two hidden layers of 16 and 32 neurons each, respectively and *tanh* activation. The GalNODE have been designed with one hidden layer of 32 neurons whose depth–varying weights were parametrized by a Fourier series of five modes. The resulting trajectories over the learned vector fields are shown in Fig. 10.

**Data–controlled Neural ODEs**   We evaluate both the handcrafted linear depth–invariant model (9) and the general formulation of data–controlled models (10), realized with two hidden layers of 32 neurons each and *tanh* activation in all layers but the output. Note that the loss of the handcrafted model results to be convex and continuously differentiable. Moreover, proof A.5 provides analytically a lower bound on the model parameter to ensure the loss to be upper–bounded by a desired $\epsilon$, making its training superfluous. Nevertheless, we provide results with a trained version to show that the benefits of data–controlled Neural ODEs are compatible with gradient–based learning.

The results are shown in Fig.s 10 and 11. The input data information embedded into the vector field allows the Neural ODE to steer the hidden state towards the desired label through its continuous depth. Data–controlled Neural ODEs can be used to learn challenging maps (Dupont et al., 2019) without augmentation.

**Concentric annuli with non–augmented variants**   We train each model for 1024 iterations using AdamW with learning rate $10^{-3}$, weight decay $10^{-6}$ and batch size 1024. All models have a single hidden layer of dimension 32. The GalNODE layer is parametrized by a Fourier series of five modes.

**Conditional continuous normalizing flows**   We train data–controlled continuous normalizing flows for 2000 iterations with samples of size $2^{14}$. We use AdamW with learning rate $10^{-3}$ and weight decay $10^{-7}$. Absolute and relative tolerances of the chosen solver, dopri5 are set to $10^{-8}$. The CNF network have 2 hidden layers of dimension 128 with softplus nonlinearities.

**Adaptive depth Neural ODEs**   The experiments have been carried out with a depth–variant Neural ODE in "concat" style where $f$ was parametrized by a neural network with two hidden layers of 8 units and tanh activation. Moreover, the function $g_\omega(\mathbf{x})$ computing the data–adaptive depth of

Figure 11: Depth evolution over the learned vector fields of (9) and a data–controlled Neural ODE. As discussed in Sec.5 introducing data–control allows the model to approximate the map $\varphi(x) = -x$.

Figure 12: Evolution of the input data through the depth of the Neural ODEs

the Neural ODE was composed by a neural network with one hidden layer (8 neurons and ReLU activation) whose output is summed to one and then taken in absolute value,

$$g(\mathbf{x}) = \left| 1 + \mathbf{w}_o^\top \sigma(\mathbf{w}_i \mathbf{x} + \mathbf{b}_i) + b_o \right|$$

where $\sigma$ is the ReLU activation, $\mathbf{w}_o$, $\mathbf{w}_i$, $\mathbf{b}_i \in \mathbb{R}^8$ and $\omega = (\mathbf{w}_o, b_o, \mathbf{w}_i, \mathbf{b}_i)$. In particular, the summation to one has been employed to help the network "sparsify" the learned integration depths and avoid highly stiff vector fields, while the absolute value is needed to avoid infeasible integration intervals. The training results can be visualized in Fig. 12. This early result should be intended as a proof of concept rather than a definitive evaluation of the depth adaptation methods, which we reserve for future work. We note that the result of Fig. 5 showed in the main text has been obtained by training the model only on $x \in \{-1, 1\}$ and manually setting $s^*_{-1} = 1$, $s^*_1 = 3$.