[Reviews · NeurIPS 2020]

Review 1

Summary and Contributions: This paper provides a very thorough theoretical and empirical investigation into several aspects of neural ODEs, including both representational power, training dynamics and higher order behavior. The paper is split into 3 substantial contributions: 1. “Depth variance”: parameterizing the weights of the ODE function to also depend on depth. 2. “Augmentation strategies”: extending augmentation strategies both to input dependent augmentation and higher order dynamics. 3. “Adaptive depth and data-controlled flows”: predicting the final integration time with an extra learned network and parameterizing the ODE function as a family of vector fields conditioned on the input data. In more detail: 1. The authors show (as also discussed by Zhang et al. and others) that Neural ODEs are effectively the limit of ResNets with constant weights theta (as the time parameter appears only as an input to the ODE function f_theta and doesn’t directly parameterize different sets of thetas for different times). The authors consider two methods to tackle this weight depth dependence: expanding the weights in an orthogonal function basis and learning the basis coefficients, and learning piecewise constant weights (effectively corresponding to stacking several neural ODEs). The paper also introduces a generalized adjoint method, which extends the adjoint method from the original neural ODE paper to cases where the losses depend on the entire domain of integration and not just the final time. They show through experiments that the proposed approach can help, in particular in the extrapolation regime and for overcoming some of the representational limitations of standard neural ODEs. 2. The authors show that the augmentation strategy proposed by Dupont et al. can be generalized by making the augmentation input dependent. They further show that these augmentation strategies can also be interpreted as higher order ODEs and specifically construct examples of these. They show through experiments on MNIST that this can improve performance over naive zero augmentation. 3. The authors focus on different methods to overcome the representational limitations of neural ODEs that do not involve augmentation. The first is data controlled ODEs which parameterize a family of vector fields that depend on the input as opposed to a single vector field. The authors show that this provides an elegant way for overcoming some of the tasks proposed by Dupont et al. They also propose another surprising and original method for overcoming these limitations by learning the end time of the integration limits with another neural network. -------------------------- Post rebuttal: Thank you for the thorough and clear rebuttal. It is great to see that the proposed augmentation approaches also improve performance on CIFAR10. I somewhat agree with other reviewers that the large number of approaches explored in this paper mean that each experiment on its own can seem small. However, I do not think this is a major problem as the experiments still provide very interesting insights and future work can explore these different approaches in more depth if needed. I also appreciate the inclusion of new CIFAR10 experiments which provide further empirical evidence for the value of the proposed approaches. Overall I still think this is a strong paper and would be valuable to the NeurIPS community. Based on this I've decided to keep my score as it is and I think the paper deserves to be accepted.

Strengths: Strengths: - The paper is extremely thorough. The authors consider a very large number of experiments and settings and investigate each of them in depth - The paper provides a nice mix of both theoretical and empirical insights. Each approach is justified both from a theoretical and empirical point of view, which is nice. - The presentation and visualizations are very good and help make the paper very clear. Significance: - Neural ODEs are becoming an increasingly common model family used in machine learning and this paper provides an important and significant analysis of some of the behaviours of these models. - I expect the insights from this paper to be relevant to anyone working on neural ODEs Novelty: - The insights and experiments related to the limitations and understanding of neural ODEs are interesting and original - The proposed approaches for overcoming these limitations are novel (although other authors have discussed both depth variance and different augmentation strategies before, although not in as much depth)

Weaknesses: Some (minor) weaknesses: - The image experiments for different augmentations are only run on MNIST. Given that all the code is already set up, I think it should be relatively easy to run this also on e.g. CIFAR10. Since the difference in performance is usually bigger on this dataset, it would be interesting to see if the augmentation strategies help more in this case. - It would also be nice to see a discussion of which aspects of the improvements proposed in the paper could be relevant for continuous normalizing flows. For example, augmentation is in general hard to use in the context of CNFs since it requires marginalizing out the added dimensions. It would be interesting if you could comment on how some of the improvements you proposed could be used in this context (e.g. depth variance, data control, adaptive depth and so on).

Correctness: The claims and methods proposed in the paper are, to the best of my understanding, correct. The empirical methodology is thorough and correct too. I appreciate that numerical results are given with means and standard deviations over several runs. I have only skimmed the proofs in the appendix and they all appear correct, although I have not checked this in detail. The authors also submitted the code to reproduce all the experiments in the paper which is great.

Clarity: The paper is very well written and all methods are clearly motivated and explained. It is clear to see that the authors have put in a lot of effort to make the paper as clear as possible and it is a pleasure to read. I also enjoyed the color coding the authors use to refer to different aspects of the paper (e.g. purple for depth variance, blue for augmentation strategies etc). The visualizations are also outstanding and provide a very good intuition for many of the results. Some small clarity issues: - Why do you use s as the time variable? Wouldn’t it be easier to use t as is standard? - Some language is unnecessarily formal, e.g. in line 51 it is clear that K is a set of indices, there is no need to call K “a linearly-ordered finite subset of N”. The paragraph on well-posedness is also a little heavy.

Relation to Prior Work: The paper clearly discusses how the proposed methods and insights are related to previous works. Even though I feel I have a good knowledge of the field of this paper, I was happy to see new relevant references I didn’t know about. The related work and literature review is definitely very thorough and a strong point of this paper.

Reproducibility: Yes

Additional Feedback: Typo line 444 in appendix: gradeints -> gradients Typo line 550 in appendix: “is used extensively” is repeated


Review 2

Summary and Contributions: This paper studies a range of modifications to Neural ODEs and the effects of these modifications on toy problems. 1) The authors propose to parameterize neural dynamics with time/depth dependent parameters. They derive the analytic form of the gradient-descent in function space for these time dependent parameters and implement their method by using a finite eigenbasis approximation for the function. The basis expansion of the parameter function, when chosen approriately for the data, provides neural ODEs with inductive biases such as periodicity, allowing them to extrapolate well on sinusoids. 2) the authors propose two ways to augment the dimension of a neural ODE. Working in a higher dimensional latent space was previously shown by Dupont et al., 2019 to be beneficial for learning, but Dupont et al., 2019 used a specific linear transformation. Instead, the authors propose to use a learned linear transformation. Alternatively, the authors propose to augment a subset of the dimensions to encode a higher-order neural ODE without adding signifcant parameters. Empirically, the authors show that these perform similarly to the augmented neural ODE (ANODE) by Dupont et al., 2019 on MNIST. Compared to ANODE, the model with a learned linear transformation required ~25% fewer function evaluations per forward pass while the higher-order required ~20% more evaluations. 3) the authors propose to make neural dynamics in the latent space depend on the input data, essentially introducing a skip connection from the first timestep to each future ODE update. To motivate this modification, the authors show existence of a ODE with data dependent dynamics that can learn the reflection function which cannot be learned by vanilla neural ODEs, shown by Dupont et al., 2019. The authors show that such "data-controlled neural ODEs" can learn a good decision boundary for binary classification of concentric spheres. 4) the authors propose to learn the upper integration limit with a separate neural network and derive the gradient equations for this separate network. 5) the authors include additional case studies such as the redundancy of using a non-linear mapping to a high dimensional space, regularization, and the effects of activation function choices on neural ODEs (in the appendix). ************************** Post rebuttal: Thanks for correcting my misunderstanding about the sinusoids example and adding additional evaluations for the proposed augmentations. The authors addressed most of my concerns in their rebuttal and I am raising my score to a 6.

Strengths: - The authors propose and evaluate a large number of modifications to vanilla neural ODEs, including novel derivations such as function space gradients for time/depth dependent parameters and the gradients of a network that predicts the integration limit - Although an integrated loss function has appeared in related works, the authors present a streamlined derivation of the adjoint equations for an integrated loss function. - The authors show results on design choices that are often left out in the literature, such as the effects of activation functions and choice of input network. - Input dependent neural dynamics was well motivated by a toy example that can learn the reflection function. - The proposed methods perform competitively with augmented neural ODEs on the evaluated datasets - The experiments were targeted to show the effects of each modification and avoids confounding factors - The derivations provided in the appendix are complete and correct - The paper is well-written and easy to follow.

Weaknesses: The evaluation of each method is the main weakness. Evaluating the modifications on controllable toy data is great, but including more interesting controlled scenarios would make the paper much stronger. The authors may have stretched themselves too thin by exploring too many modifications to neural ODEs in one paper. Time/depth dependent parameters: - The empirical evaluation of Galerkin neural ODEs is limited to learning sine curves using parameter functions that are also sine curves (truncated Fourier basis). It's unclear how Galerkin neural ODEs or stacked neural ODEs will perform for more complex functions where the choice of basis is less obvious. For example, can you learn sin(x)/x which doesn't have a finite Fourier expansion? Showing more complex situations where Galerkin or stacked neural networks are beneficial would make the paper much stronger. Neural ODE augmentation schemes: - The augmentation schemes proposed are simple, which is okay if the schemes are effective or well-analyzed. However the evaluation was limited to testing/training curves, test accuracy, NFE, and #params on MNIST, producing similar to ANODEs. More analysis would make the paper stronger. For example, do higher-order augmentations make neural ODEs more expressive or easier to optimize? Why does a learned linear transform produce similar results to an identity transform? etc. Data-controlled neural ODEs/depth-adaptation: - Although the authors showed that making the dynamics input dependent allows models to learn the reflection function, it is unclear if input-dependent dynamics offer any realistic benefits on more complex problems. For example, how do they compare to ANODEs or vanilla NODEs in realistic tasks beyond the reflection function? Similarly, empirical evaluation of depth-adapted neural ODEs was limited to showing that they can learn the reflection map.

Correctness: Claims and methodology are correct.

Clarity: The paper is easy to follow and well written.

Relation to Prior Work: Yes prior work is appropriately discussed. The authors may want to point out that prior works have also considered the integrated loss function. For example https://arxiv.org/abs/1909.12077 and https://arxiv.org/abs/2002.12880

Reproducibility: Yes

Additional Feedback: Equation 3: "\dot{z}" → "\dot{z}(s)" Line 78: "This approach" Is this referring to the hypernetwork approach? Line 91: "to gradient the loss gradient" → "to compute the loss gradient" Line 104: "Appendix B" → Appendix C.1" Figure 1: What are the blue curves? Are they training data with different initial conditions? Why does the variance decrease over time? Line 117: Missing reference "in Appendix" Figure 2: Is each blue curve a component of theta(s) over time/depth s? Line 195: "h(1)" → "z(1)" Figure 6: May want to clarify that "Depth-variant" corresponds to the concatenation scheme from Chen et al., 2018 "Trajectory tracking" subsection of the appendix: What was the neural network architecture for the Galerkin NODE? Also number of training points and batch size? "Depth- varying classification" subsection of the appendix: "Architecture, number of training points, and batch size?" Figure 11 in appendix: Title "Time invariant" → "Depth invariant" to be consistent with the paper Figure 15 in appendix: Caption is the same as Figure 14


Review 3

Summary and Contributions: This paper studies several extensions of the original neural ODE model from interesting angles, such as alternative designs of a time-varying parameter and augmentation, and data-dependent vector fields/model depth.

Strengths: The paper contains several interesting ideas such as a data-dependent model depth design. The investigation of these ideas, although still somewhat preliminary, may lead to more impactful future developments.

Weaknesses: Apart from a few technical ambiguities and flaws, the main limitation of this paper is that it attempts to discuss many modifications at the same time. While this itself is never a problem, the paper could be further strengthen significantly by giving more direct advice to practitioners on when to choose what.

Correctness: The claims are mostly correct. Here're a few comments/questions I have: 1. In table 1, the "second-order" column seems to have slightly worse test accuracy, NFE, and roughly the same number of parameters. This is somewhat contradictory to the "more param. efficient" argument given in the higher-order node section. Is there any analysis (either theory/empirical) on why this is the case in practice? 2. In section 5, it is initially mentioned that "it is indeed true that two distinct trajectories can never intersect in the state-space in the one-dimensional case". Given the analysis in section 5.2, this might actually be possible, if different trajectories could travel for different amounts of time. 3. In figure 2 bottom row, why are there multiple trajectories of theta?

Clarity: The paper is mostly well-written, with the exception that it's unclear how theorem 1 relates to the rest of the analysis in the paper.

Relation to Prior Work: Several additional closely related works should be mentioned. [1] is one of the early important works that studies the connection between DEs and ResNets. [2, 3] are important works linking control, dynamical systems, and machine learning. [4] is an important work on extending the framework to the stochastic case. The latent SDE model used in sections 7.2 and 7.3 in their experiments also has the element of using a data-dependent vector field, since the drift function there is different for each time series trajectory, and the cost is amortized by letting the drift function take in an additional context produced by a GRU. [1] Lu, Yiping, et al. "Beyond finite layer neural networks: Bridging deep architectures and numerical differential equations." International Conference on Machine Learning. 2018. [2] Weinan, E. "A proposal on machine learning via dynamical systems." Communications in Mathematics and Statistics 5.1 (2017): 1-11. [3] Li, Qianxiao, et al. "Maximum principle based algorithms for deep learning." The Journal of Machine Learning Research 18.1 (2017): 5998-6026. [4] Li, Xuechen, Ting-Kam Leonard Wong, Ricky TQ Chen, and David Duvenaud. "Scalable gradients for stochastic differential equations." arXiv preprint arXiv:2001.01328 (2020).

Reproducibility: Yes

Additional Feedback: About theorems 1 and 2: These are statements that are straightforward to derive (and can be derived in a few lines) from the original Pontryagin maximum principle (PMP), which is a standard textbook result in introductory control theory books. I wouldn't give them such a big title as "theorems" if I were the author. I think the title "proposition" is more appropriate. In addition, the relation between the statement and PMP should be clarified in the text. The "curse of dimensionality" (COD) that I am aware of mainly refers to the situation where high-dimensional spaces have properties that are drastically different from lower dimensional spaces (e.g. 1d, 2d). The main cause of this is the expansion of volume. It is unclear what "exponential increase in complexity" means on line 179. In addition, I don't think what's described at the top of section 5 is really due to COD. Minor: - Writing the time integral as \int_\mathcal{S} can be a little confusing, since integrating from t0 to t1 isn't the same as integrating from t1 to t0. - line 171: "this does necessarily hold in general". Do the authors mean the opposite? - line 97: "is turned into final dimension". Should this be "finite dimension"? - line 121: "proposed approach of the seminal paper". "Seminal paper" is probably too strong a description. I am very willing to modify the "overall score" if the authors can address my concerns in the "Correctness" and "Relation to prior work" sections, and the current section of the review. Post-rebuttal: The authors have addressed some of my concerns regarding experimentation, so I think the quality of the empirical aspect of the paper is better. With this, I'm raising my score to a 7. I still think the theorem statements are a bit too bold. Theorem 2 perhaps slightly deviates away from PMP in some sense, though it should straightforward to derive with a little bit of variational calculus. Additional minor comments: The typical terminology on this topic is "function space" not "functional space". In analysis, the term "functional" typically means a mapping which takes an element in function space to a real/complex number.


Review 4

Summary and Contributions: The paper proposes several modifications to the framework of NeuralODE. It presents changes at the input level, at the ODE level, in the augmentation strategies, as well as some conditional variants. It shows the effectiveness of these methods in improving representational capacity.

Strengths: The paper present various strategies that are theoretically sound. Taken together, these offer several simple modifications that are quite effective in improving the use of Neural ODEs beyond the original formulation. It provides helpful visualizations as well. In addition, it provides a conditional variant that would be helpful in modelling conditional computations well.

Weaknesses: The effectiveness is displayed first theoretically, and then only on simpler use cases. It would be great to show their effectiveness in more complex use cases, such as continuous normalizing flows, as mentioned in Section 6. This would be akin to making a conditional generative model out of Neural ODEs.

Correctness: The claims are supported by relevant experiments building from previous work. Since the paper covers several proposals, it is limited in experimentation, but sufficient to show each method's effectiveness.

Clarity: The paper is well-written, it provided good insights into how to expand the original Neural ODE formulation in several ways, and is organized well.

Relation to Prior Work: Yes, previous works are clearly mentioned along with the incremental changes from the original formulation.

Reproducibility: Yes

Additional Feedback: -------------------------------- I believe the reviewers' concerns were duly addressed in the rebuttal. I am increasing my rating by 1. I believe this is a strong paper that should be accepted at NeurIPS 2020.

[Author Response · NeurIPS 2020]

We thank the reviewers for valuable and thoughtful feedback, and for acknowledging the importance of this work.

**Scope:** Our main objective has been to provide insights on current limitations of the Neural ODE framework, and to

design novel solutions backed by theory. Although evaluations for specific use cases e.g. CNF was not in within our

primary scope, we include additional results to address requests and provide further evidence of practical usefulness.

**Further augmentation evaluations [R2, R3]:** We obtained improved results on all variants by means of a similar

architecture equipped with a pooling layer (closer to standard image class. approaches) and tolerances $10^4$. The neural

ODEs learn richer dynamics (higher NFEs), boosting performance across all models and clarifying the relative ranking

of aug. approaches. This also leads to improved parameter efficiency ($\approx 8$x less parameters). Tab. 1 includes results

on both MNIST as well as CIFAR10 **(R1)**. The parameter efficiency of $2^{nd}$–order models **(R2, R3)** is now more

pronounced. We also report that they converge faster, often several epochs ahead of the alternatives.

**Discussion on CNFs [R1, R2, R4]:** We further showcase *data-control* (DC) strategies in the context CNFs as a more

complex task. Compared to regular CNFs, DC–CNFs do not require changes to the formulation and converge faster

with simpler flows as shown in Fig. 1, effectively reducing NFEs. Adaptive–depth models are also compatible with

CNFs and would allow the model to allocate more *depth* to data further away from their target destination.

**Depth-variance techniques [R2]:** We agree that the choice of basis in Galërkin neural ODEs is important and worthy

of several standalone investigations. However, regarding the *sinusoids* example, periodicity of the weights (inferred by

the choice of the Fourier eigenbasis) does not imply periodicity of the Neural ODE and hence does not constitute a

strong inductive bias. To confirm this, we tested with different signals and eigenbasis (Chebychev poly., RBFs). Fig. 2

shows a more complex experiment for time–varying nonlinear system.

**Signal tracking [R2]:** Here, depth–variance is not needed to actually learn the trajectory, generated by $\ddot{x} =$

$x, \; [x_0, \dot{x}_0] = [1, 0]$ which does not contain any depth-varying harmonics. Rather, it ensures that for any initial

condition of the neural ODE, sampled from $\mathcal{N}([1, 0], \sigma)$, the solution converges to the signal. The same result can be

obtained for nonlinear systems whose solution does not admit a finite spectral decomposition as shown above.

**Related work [R3]:** We agree that these important references belong to Section 6 and have made the suggested changes.

The approach of latent neural SDEs (and ODEs) is different compared to data-control, which does not require variational

inference. It is correct to state that, however, both approaches condition the vector field on data.

**Relation to PMP [R3]:** We agree that Th.m 1 is directly derived via classic optimal control theory (PMP) and should

be more appropriately referred to as "Proposition". However, including it was necessary for two main reasons: to extend

vanilla adjoints to integral loss functions (used in practice for CNFs or signal tracking but not yet implemented) and to

set the stage for Th.m 2. We note that Th.m 2 is a non–trivial generalization to infinite dimensional spaces.

**Relation to COD [R3]:** We agree on COD and clarified the statements in Sec. 5. The phenomenon we want to highlight

is that dimension of the state–space also drastically affects the behavior of dynamical systems in general.

**Guidelines on choosing correct variants [R3]:** In general, we observe data–control to be beneficial in all settings. We

agree that additional guidelines on model choice could be useful to the reader; we will add more information.

**Given Sec. 5.2, state–space crossing might be possible if each traj. could travel for different amounts of time.**

[R2]:In 5.2 we argue that adaptive–depth models can learn the reflection map **without** crossing flows (as they still

cannot cross), consistently to what is stated in rest of Sec. 5. This is in fact the main *leitmotiv* of adaptive-depth models.

**Clarifications: Figure 1 [R2]:** The blue curves are learned flows of test init. cond., which converge to the signal

to track (hence the decreasing variance across *depth*). **Figure 2 [R2, R3]:** Each traj. represents the evolution of a

single parameter. **Training details [R2]** On the signal tracking task, we train on $10^2$ initial conditions, full batch. The

GalNODE architecture has a hidden layer of 64. On depth-varying classification, the architectures have two hidden

layers of 32 with a dataset of $10^4$ points (dense, to approx. *connected annuli*). **Why $s$ instead of $t$? [R1]:** We chose $s$

against $t$ as a more general formulation for the (continuous) depth, to avoid confusion in static settings or whenever

time is not directly related to depth–propagation dimension. **"This approach" refers to hypernetworks? [R2]:** Yes.

**Typos details [R1, R2, R3]** : We addressed all the remaining typos.

Figure 1: Data–controlled (DC) CNF vs vanilla CNF (3 Gaussians). The proposed
model converges faster to a better solution and with simpler flows (thus lower NFEs).

Figure 2: Galërkin Neural ODEs with Chebychev eigenbasis trained with integral loss to track the nonlinear time-varying *Duffing oscillator* $\ddot{x} = -\alpha x(1 + x^2) + \beta \cos(\omega s)$ ($\alpha = .5, \beta = 3, \omega = 5$). All traj.s, starting from rand. ICs, converge to the desired signal.

| | NODE | | ANODE | | IL-NODE | | 2nd–Ord. | |
|---|---|---|---|---|---|---|---|---|
| | MNIST | CIFAR | MNIST | CIFAR | MNIST | CIFAR | MNIST | CIFAR |
| Test Acc. | 98.3 | 59.1 | 99.1 | 68.7 | 99.4 | 70.7 | 99.5 | 71.8 |
| NFE | 130 | 152 | 124 | 153 | 112 | 150 | 106 | 142 |
| Param.[K] | 6.4 | 42.1 | 6.4 | 41.4 | 6.4 | 41.9 | 5.8 | 37.4 |

Table 1: Test results, 5 runs on MNIST, 3 on CIFAR10* (final results might vary slightly.)

[Meta-Review · NeurIPS 2020]

The four expert reviewers all agree that this paper contributes not just one but several noteworthy new insights on neural ODEs, a topic of enormous ongoing attention in the community. A clear accept.